# Training Feedback Spiking Neural Networks by Implicit Differentiation on the Equilibrium State

**Mingqing Xiao**[1], **Qingyan Meng**[2,3], **Zongpeng Zhang**[4], **Yisen Wang**[1,5], **Zhouchen Lin**[1,5,6]*

[1]Key Laboratory of Machine Perception (MoE), School of AI, Peking University
[2]The Chinese University of Hong Kong, Shenzhen
[3]Shenzhen Research Institute of Big Data
[4]Center for Data Science, Academy for Advanced Interdisciplinary Studies, Peking University
[5]Institute for Artificial Intelligence, Peking University
[6]Pazhou Lab, Guangzhou 510330, China
{mingqing_xiao, yisen.wang, zlin}@pku.edu.cn, qingyanmeng@link.cuhk.edu.cn,
zongpeng.zhang98@gmail.com

## Abstract

Spiking neural networks (SNNs) are brain-inspired models that enable energy-efficient implementation on neuromorphic hardware. However, the supervised training of SNNs remains a hard problem due to the discontinuity of the spiking neuron model. Most existing methods imitate the backpropagation framework and feedforward architectures for artificial neural networks, and use surrogate derivatives or compute gradients with respect to the spiking time to deal with the problem. These approaches either accumulate approximation errors or only propagate information limitedly through existing spikes, and usually require information propagation along time steps with large memory costs and biological implausibility. In this work, we consider feedback spiking neural networks, which are more brain-like, and propose a novel training method that does not rely on the exact reverse of the forward computation. First, we show that the average firing rates of SNNs with feedback connections would gradually evolve to an equilibrium state along time, which follows a fixed-point equation. Then by viewing the forward computation of feedback SNNs as a black-box solver for this equation, and leveraging the implicit differentiation on the equation, we can compute the gradient for parameters without considering the exact forward procedure. In this way, the forward and backward procedures are decoupled and therefore the problem of non-differentiable spiking functions is avoided. We also briefly discuss the biological plausibility of implicit differentiation, which only requires computing another equilibrium. Extensive experiments on MNIST, Fashion-MNIST, N-MNIST, CIFAR-10, and CIFAR-100 demonstrate the superior performance of our method for feedback models with fewer neurons and parameters in a small number of time steps. Our code is available at https://github.com/pkuxmq/IDE-FSNN.

## 1  Introduction

Spiking neural networks (SNNs) have gained increasing attention recently due to their inherent energy-efficient computation [21, 38, 41, 33, 8]. Inspired by the neurons in the human brain, biologically plausible SNNs transmit spikes between neurons, enabling event-based computation which can be carried out on neuromorphic chips with less energy consumption [1, 7, 30, 33]. Meanwhile, SNNs are computationally more powerful than artificial neural networks (ANNs) theoretically and are therefore regarded as the third generation of neural network models [24].

---

*Corresponding author.

35th Conference on Neural Information Processing Systems (NeurIPS 2021).

Despite the advantages, directly supervised training of SNNs remains a hard problem, which hampers the real applications of SNNs compared with popular ANNs. The main obstacle lies in the complex spiking neuron model. While backpropagation [35] works well for ANNs, it suffers from the discontinuity of spike generation which is non-differentiable in SNN training. Most recent successful SNN training methods still imitate the backpropagation through time (BPTT) [39] framework by error propagation through the computational graph unfolded along time steps, and they deal with the spiking function by applying surrogate derivatives to approximate the gradients [41, 5, 14, 38, 42, 27, 16, 47], or by computing the gradients with respect to the spiking time only on the spiking neurons [6, 46, 16]. However, these methods either accumulate approximation error along time steps, or suffer from the "dead neuron" problem [38], i.e. learning would not occur when no neuron spikes. At the same time, BPTT requires memorizing intermediate variables at all time steps and backpropagating along them, which is memory-costing and biologically implausible. So it is necessary to consider training methods other than backpropagation along computational graphs that fit SNNs better.

On the other hand, most recent SNN models simply imitate the feedforward architectures of ANNs [41, 38, 42, 46, 47], which ignores the ubiquitous feedback connections in the human brain. Feedback (recurrent) circuits are critical to human's vision system for object recognition [15]. Meanwhile, [18] shows that shallow ANNs with recurrence achieve higher functional fidelity of human brains and similarly high performance on large-scale vision recognition tasks, compared with deep ANNs. So incorporating feedback connections enables neural networks to be shallower, more efficient, and more brain-like. As for SNNs, feedback was popular in early models like Liquid State Machine [25], which leverages a recurrent reservoir layer with weights fixed or trained by unsupervised methods. And compared with the uneconomical cost for ANNs to incorporate feedback connections by unfolding along time, SNNs naturally compute with multiple time steps, which inherently supports feedback connections. Most recent SNN models imitate feedforward architectures because they were once lacking effective training methods and thus they borrow everything from successful ANNs. We focus on another direction, i.e. feedback SNN, which is a natural choice for visual tasks as well.

In this work, we consider the training of feedback spiking neural networks (FSNN), and propose a novel method based on the Implicit Differentiation on the Equilibrium state (IDE). Inspired by recent advances in implicit models [3, 4], which treat weight-tied ANNs as solving a fixed-point equilibrium equation and propose alternative implicit models defined by the equation, we derive that when the average inputs converge to an equilibrium, the average firing rates of FSNNs would gradually evolve to an equilibrium state along time, which follows a fixed-point equation as well. Then we view the forward computation of FSNN as a black-box solver for the fixed-point equation, and borrow the idea of implicit differentiation from implicit models [3, 4] to calculate the gradients, which only relies on the equation rather than the exact forward procedure. In this way, gradient calculation is agnostic to the spiking function in SNN, thus avoiding the common difficulties in SNN training. While implicit differentiation may seem too abstract to be computed in the brain, we briefly discuss the biological plausibility and show that it only requires computing another equilibrium along the inverse connections of neurons. Besides, we incorporate the multi-layer structure into the feedback model for better representation ability. Our contributions include:

1. We are the first to theoretically derive the equilibrium states with a fixed-point equation for the average firing rates of FSNNs with the (leaky) integrate and fire model under both continuous and discrete views. According to this, the forward computation of FSNNs can be interpreted as solving a fixed-point equation.

2. We propose a novel training method for FSNNs based on the implicit differentiation on the equilibrium state, which is decoupled from the forward computational graph and avoids SNN training problems, e.g. non-differentiability and large memory costs. We also discuss the biological plausibility and demonstrate the connection to the Hebbian learning rule.

3. We conduct extensive experiments on MNIST, Fashion-MNIST, N-MNIST, CIFAR-10, and CIFAR-100, which demonstrate the superior results of our methods with fewer neurons and parameters in a small number of time steps for both static images and neuromorphic inputs. Especially, our directly trained model can outperform the state-of-the-art SNN performance on the complex CIFAR-100 dataset with only 30 time steps.

## 2 Related Work

**Training Methods for Spiking Neural Networks.** Early works apply biologically inspired method, spike-time dependent plasticity (STDP) [9], to formulate a bottom-up unsupervised learning rule, or

choose reward-modulated STDP [22] with limited supervision. Since the rise of successful ANNs, error backpropagation and gradient descent have inspired many methods. One direction is to convert a trained ANN to SNN [13, 34, 37, 32, 8, 44]. However, they suffer from conversion errors and extremely large simulation time steps. The other methods are to directly calculate the gradient and train SNNs. These methods either compute the gradient with respect to spiking times [6, 46], or leverage a surrogate derivative for discontinuous spiking functions [21, 41, 5, 14, 38, 42, 27, 47], or combine them [16]. However, they suffer from the "dead neuron" problem [38] or accumulated approximation error, and typically require backpropagation along the computational graph to be unfolded by many time steps, which is memory-consuming and biologically implausible. As for SNN with feedback connection, [45] proposes the ST-RSBP method, which backpropagates errors at the spike-train level. They simply approximate the activation function of a neuron as a linear operation, and require long time steps for satisfactory results. In this work, we propose a new training method that does not rely on the exact reverse of the forward computation, which avoids problems of non-differentiability and large memory costs, and only requires short time steps for superior performance. There are also works trying methods other than BP along computational graphs to train SNNs, such as equilibrium propagation [29]. However, [29] defines a complex computation form rather than common SNN models, and can hardly achieve satisfactory results on the simple MNIST dataset. Instead, our work is based on SNN models applicable on neuromorphic hardware and demonstrates promising results on more complex datasets.

**Equilibrium of Neural Networks.** The study on the equilibrium of neural networks originates from energy-based models, e.g. Hopfield Network [11, 12]. They view the dynamics or iterative procedures of feedback (recurrent) neural networks as minimizing an energy function, which will converge to a minimum of the energy. Based on the energy, several training methods are proposed, including recurrent backpropagation [2, 31] and more recent equilibrium propagation (EP) [36]. They heavily rely on the energy function and can hardly achieve competitive results as deep neural networks do. Deep equilibrium models [3, 4], on the other hand, are recently proposed models which treat weight-tied deep ANNs as solving a fixed-point equilibrium point, and propose alternative implicit models defined by the fixed-point equations rather than energy functions. They express the entire deep network as an equilibrium computation and solve and train it by root-finding methods and implicit differentiation, respectively, which achieves superior results. Most of these works are based on ANNs, except that [29] generalizes the EP method to neurons with binary communications. They define a complex neuron computation form and follow the methodology of energy-based EP method to approximate the gradients. Several recent works also study the equilibrium of SNNs [23, 26]. They consider equilibrium from the perspective of solving a constrained optimization problem, but either do not propose to guide the training based on it or are limited in single-layer structure and simple problems. Differently, we are the first to derive the equilibrium state with a fixed-point equation for average firing rates of FSNNs with common SNN models, and propose to train SNNs by exact gradients through implicit differentiation, which is also scalable to multi-layer structure and deep learning problems.

## 3 Preliminaries

### 3.1 Spiking Neural Network Models

Spiking neurons, different from real-valued ANNs, communicate with each other by binary spike trains along time. Each neuron maintains a membrane potential, which integrates input spike trains, and the neuron would generate an output spike once the membrane potential exceeds a threshold. The commonly used integrate and fire (IF) model and leaky integrate and fire (LIF) model describe the dynamics of the membrane potential as:

$$
\begin{aligned}
\textbf{IF:} \quad & \frac{du}{dt} = R \cdot I(t), & u < V_{th}, \\
\textbf{LIF:} \quad & \tau_m \frac{du}{dt} = -(u - u_{rest}) + R \cdot I(t), & u < V_{th},
\end{aligned}
\tag{1}
$$

where $u$ is the membrane potential, $I$ is the input current, $V_{th}$ is the spiking threshold, and $R$ and $\tau_m$ are resistance and time constant, respectively. Once $u$ reaches $V_{th}$ at time $t^f$, a spike is generated and $u$ is reset to the resting potential $u = u_{rest}$, which is usually taken as 0. The spike train is expressed by the Dirac delta function: $s(t) = \sum_{t^f} \delta(t - t^f)$. We consider the simple current model

$I_i(t) = \sum_j w_{ij} s_j(t) + b$, where $w_{ij}$ is the weight from neuron $j$ to neuron $i$, which is the same as ANN. After discretization, the general computation form for the IF and LIF model is described as:

$$
\begin{cases}
u_i\,[t + 0.5] = \lambda u_i[t] + \displaystyle\sum_j w_{ij} s_j[t] + b, \\
s_i[t + 1] = H(u_i\,[t + 0.5] - V_{th}), \\
u_i[t + 1] = u_i\,[t + 0.5] - V_{th} s_i[t + 1],
\end{cases}
\tag{2}
$$

where $H(x)$ is the Heaviside step function, i.e. the non-differentiable spiking function, $s_i[t]$ is the binary spike train of neuron $i$, and $\lambda$ is 1 for the IF model while $\lambda < 1$ is a leaky term related to the constant $\tau_m$ and discretization time interval for the LIF model. The constant $R$, $\tau_m$, and time step size are absorbed into the weights $w_{ij}$ and bias $b$. We use subtraction as the reset operation.

## 3.2 Implicit Differentiation on the Fixed-Point Equation

We consider a fixed-point equation $\mathbf{a} = f_\theta(\mathbf{a})$ parameterized by $\theta$. Let $\mathcal{L}(\mathbf{a}^*)$ denote the objective function with respect to the equilibrium state $\mathbf{a}^*$, and let $g_\theta(\mathbf{a}) = f_\theta(\mathbf{a}) - \mathbf{a}$. The implicit differentiation on the equation satisfies $\left(I - \frac{\partial f_\theta(\mathbf{a}^*)}{\partial \mathbf{a}^*}\right) \frac{\mathrm{d}\mathbf{a}^*}{\mathrm{d}\theta} = \frac{\partial f_\theta(\mathbf{a}^*)}{\partial \theta}$ [3]. Therefore, the differentiation of $\mathcal{L}(\mathbf{a}^*)$ with respect to parameters can be calculated based on implicit differentiation:

$$
\frac{\partial \mathcal{L}(\mathbf{a}^*)}{\partial \theta} = -\frac{\partial \mathcal{L}(\mathbf{a}^*)}{\partial \mathbf{a}^*} \left(J_{g_\theta}^{-1}|_{\mathbf{a}^*}\right) \frac{\partial f_\theta(\mathbf{a}^*)}{\partial \theta},
\tag{3}
$$

where $J_{g_\theta}^{-1}|_{\mathbf{a}^*}$ is the inverse Jacobian of $g_\theta$ evaluated at $\mathbf{a}^*$. For the proof please refer to [3]. To solve the inverse Jacobian, we follow [3, 4] and solve an alternative linear system $\left(J_{g_\theta}^T|_{\mathbf{a}^*}\right)\mathbf{x} + \left(\frac{\partial \mathcal{L}(\mathbf{a}^*)}{\partial \mathbf{a}^*}\right)^T = 0$. We can leverage Broyden's method [3, 4], which is a second-order quasi-Newton approach; or we can alternatively use a fixed-point update scheme $\mathbf{x} = (J_{f_\theta}^T|_{\mathbf{a}^*})\mathbf{x} + \left(\frac{\partial \mathcal{L}(\mathbf{a}^*)}{\partial \mathbf{a}^*}\right)^T$ since $J_{g_\theta}^T|_{\mathbf{a}^*} = J_{f_\theta}^T|_{\mathbf{a}^*} - I$, and it converges with linear convergence rate as long as $\|J_{f_\theta}^T|_{\mathbf{a}^*}\| < 1$. In this way, gradients for the parameters can be calculated only with the equilibrium state and equation.

# 4 Proposed IDE Method

In this section, we first derive the equilibrium state of FSNNs under both continuous and discrete views, and demonstrate that FSNNs can be treated as solving a fixed-point equation. Then we introduce how to train the network by the proposed IDE method based on the equation and briefly discuss the biological plausibility. Finally, we incorporate the multi-layer structure into the model for more non-linearity and stronger representation ability.

## 4.1 Derivation of Equilibrium States for Feedback Spiking Neural Networks

### 4.1.1 Continuous View

We first consider a group of spiking neurons with feedback connections. Let $\mathbf{u}(t)$ and $\mathbf{s}(t)$ denote the membrane potentials and spikes of these neurons at time $t$ respectively, $\mathbf{x}(t)$ denote the inputs, $\mathbf{W}$ denote the feedback weight matrix, $\mathbf{F}$ denote the weight matrix from inputs to these neurons, and $\mathbf{b}$ denote a constant bias. Under the IF model, the dynamics of membrane potentials are expressed as:

$$
\frac{\mathrm{d}\mathbf{u}}{\mathrm{d}t} = \mathbf{W}\mathbf{s}(t - \Delta t_d) + \mathbf{F}\mathbf{x}(t) + \mathbf{b} - V_{th}\mathbf{s}(t),
\tag{4}
$$

where $\Delta t_d$ is a time delay of feedback connections, and $V_{th}$ is the threshold. Note that $\mathbf{W}$ and $\mathbf{F}$ represent linear operations including both fully-connected and convolutional layers. Define the average firing rates as $\mathbf{a}(t) = \frac{1}{t}\int_0^t \mathbf{s}(\tau)\mathrm{d}\tau$, and the average inputs as $\overline{\mathbf{x}}(t) = \frac{1}{t}\int_0^t \mathbf{x}(\tau)\mathrm{d}\tau$. Then through integration, we have:

$$
\mathbf{a}(t) = \frac{1}{V_{th}}\left(\frac{t - \Delta t_d}{t}\mathbf{W}\mathbf{a}(t - \Delta t_d) + \mathbf{F}\overline{\mathbf{x}}(t) + \mathbf{b} - \frac{\mathbf{u}(t)}{t}\right).
\tag{5}
$$

Eq.(5) roughly follows a fixed-point update scheme except the existence of $\mathbf{u}(t)$. Now we dive into $\mathbf{u}(t)$. Since neurons will not spike when the accumulated $\mathbf{u}(t)$ is negative, if $v_i(t) = \left(\frac{t-\Delta t_d}{t}\mathbf{W}\mathbf{a}(t-\Delta t_d) + \mathbf{F}\overline{\mathbf{x}}(t) + \mathbf{b}\right)_i < 0$, ideally neuron $i$ generates no spike and its accumulated negative term remains in $\mathbf{u}_i(t)$. So $\mathbf{u}_i(t)$ can be divided as $\mathbf{u}_i(t) = \mathbf{u}_i^-(t) + \mathbf{u}_i^+(t)$, where $\frac{1}{t}\mathbf{u}_i^-(t) = \min(v_i(t), 0)$ is the remaining negative term, and $\mathbf{u}_i^+(t)$ is the positive one typically bounded in the range between $0$ and $V_{th}$. There could be random error in $\mathbf{u}_i^+(t)$ in the context of random arrival of spikes rather than the average condition (e.g. the average is $0$, but a large positive input followed by a negative one will generate an unexpected spike). Despite this, we can still assume $\mathbf{u}_i^+(t)$ to be bounded by a constant when average inputs converge. By this decomposition, we have the equation with the element-wise ReLU function ($\text{ReLU}(x) = \max(x, 0)$) and bounded $\mathbf{u}^+(t)$:

$$\mathbf{a}(t) = \text{ReLU}\left(\frac{1}{V_{th}}\left(\frac{t-\Delta t_d}{t}\mathbf{W}\mathbf{a}(t-\Delta t_d) + \mathbf{F}\overline{\mathbf{x}}(t) + \mathbf{b}\right)\right) - \frac{1}{V_{th}}\frac{\mathbf{u}^+(t)}{t}. \tag{6}$$

With Eq.(6), we can derive that the average firing rate will gradually converge to an equilibrium state.

**Theorem 1.** *If the average inputs converge to an equilibrium point $\overline{\mathbf{x}}(t) \to \mathbf{x}^*$, and there exists constant $c$ and $\gamma < 1$ such that $|\mathbf{u}_i^+(t)| \le c, \forall i, t$ and $\|\mathbf{W}\|_2 \le \gamma V_{th}$, then the average firing rates of FSNN with continuous IF model in Eq.(6) will converge to an equilibrium point $\mathbf{a}(t) \to \mathbf{a}^*$, which satisfies the fixed-point equation $\mathbf{a}^* = ReLU\left(\frac{1}{V_{th}}\left(\mathbf{W}\mathbf{a}^* + \mathbf{F}\mathbf{x}^* + \mathbf{b}\right)\right)$.*

The proof can be found in Appendix C. Theorem 1 rigorously shows the equilibrium state under the IF model, and we can view the forward computation of FSNN as solving this fixed-point equation.

As for the LIF model, we can similarly define the weighted average firing rate $\hat{\mathbf{a}}(t) = \frac{\int_0^t \kappa(t-\tau)\mathbf{s}(\tau)\mathrm{d}\tau}{\int_0^t \kappa(t-\tau)\mathrm{d}\tau}$ and the weighted average inputs $\hat{\mathbf{x}}(t) = \frac{\int_0^t \kappa(t-\tau)\mathbf{x}(\tau)\mathrm{d}\tau}{\int_0^t \kappa(t-\tau)\mathrm{d}\tau}$, where $\kappa(\tau) = \exp(-\frac{\tau}{\tau_m})$ is the response kernel of the LIF model. In this setting, however, there could be random errors caused by $\mathbf{u}^+(t)$ as its denominator $\int_0^t \kappa(t-\tau)\mathrm{d}\tau$ does not go to infinity. We consider it as an approximate solver for the equilibrium with random errors, as shown in Proposition 1. Please refer to Appendix E for details.

**Proposition 1.** *If the weighted average inputs converge to an equilibrium point $\hat{\mathbf{x}}(t) \to \mathbf{x}^*$, and there exists constant $c$ and $\gamma < 1$ such that $|\mathbf{u}_i^+(t)| \le c, \forall i, t$ and $\|\mathbf{W}\|_2 \le \gamma V_{th}$, then the weighted average firing rates $\hat{\mathbf{a}}(t)$ of FSNN with continuous LIF model gradually approximate an equilibrium point $\mathbf{a}^*$ with bounded random errors, which satisfies $\mathbf{a}^* = ReLU\left(\frac{1}{V_{th}}\left(\mathbf{W}\mathbf{a}^* + \mathbf{F}\mathbf{x}^* + \mathbf{b}\right)\right)$.*

### 4.1.2 Discrete View

In practice, we will simulate SNNs with discretization. Now consider the computation in Eq.(2). With feedback connections, the update equation of membrane potentials under the IF model is:

$$\mathbf{u}[t+1] = \mathbf{u}[t] + \mathbf{W}\mathbf{s}[t] + \mathbf{F}\mathbf{x}[t] + \mathbf{b} - V_{th}\mathbf{s}[t+1], \tag{7}$$

where we treat the feedback delay in one time step for simplicity. Define the average firing rates as $\mathbf{a}[t] = \frac{1}{t}\sum_{\tau=1}^t \mathbf{s}[\tau]$, the average inputs as $\overline{\mathbf{x}}[t] = \frac{1}{t+1}\sum_{\tau=0}^t \mathbf{x}[\tau]$, and $\mathbf{u}[0] = \mathbf{0}, \mathbf{s}[0] = \mathbf{0}$. By summation, we have:

$$\mathbf{a}[t+1] = \frac{1}{V_{th}}\left(\frac{t}{t+1}\mathbf{W}\mathbf{a}[t] + \mathbf{F}\overline{\mathbf{x}}[t] + \mathbf{b} - \frac{\mathbf{u}[t+1]}{t+1}\right). \tag{8}$$

Different from the continuous view, $\mathbf{a}[t]$ is bounded in the range of $[0, 1]$, since there could be at most $t$ spikes during $t$ time steps. Therefore, $\mathbf{u}_i[t]$ will maintain both the negative terms and the exceeded positive ones. Similarly, $\mathbf{u}_i[t] = \mathbf{u}_i^-[t] + \mathbf{u}_i^+[t]$, where $\frac{1}{t}\mathbf{u}_i^-[t] = \min\left(\max\left(v_i[t] - V_{th}, 0\right), v_i[t]\right)$ is the exceeded term, and $\mathbf{u}_i^+[t]$ is assumed to be bounded by a constant as previously indicated. Then:

$$\mathbf{a}[t+1] = \sigma\left(\frac{1}{V_{th}}\left(\frac{t}{t+1}\mathbf{W}\mathbf{a}[t] + \mathbf{F}\overline{\mathbf{x}}[t] + \mathbf{b}\right)\right) - \frac{1}{V_{th}}\frac{\mathbf{u}^+[t+1]}{t+1}, \text{where } \sigma(x) = \begin{cases} 1, & x > 1 \\ x, & 0 \le x \le 1 \\ 0, & x < 0 \end{cases}. \tag{9}$$

With Eq.(9), we can derive the equilibrium state under discrete view.

**Theorem 2.** *If the average inputs converge to an equilibrium point $\overline{\mathbf{x}}[t] \to \mathbf{x}^*$, and there exists constant $c$ and $\gamma < 1$ such that $|\mathbf{u}_i^+[t]| \le c, \forall i, t$ and $\|\mathbf{W}\|_2 \le \gamma V_{th}$, then the average firing rates of FSNN with discrete IF model in Eq.(9) will converge to an equilibrium point $\mathbf{a}[t] \to \mathbf{a}^*$, which satisfies the fixed-point equation $\mathbf{a}^* = \sigma\left(\frac{1}{V_{th}}\left(\mathbf{W}\mathbf{a}^* + \mathbf{F}\mathbf{x}^* + \mathbf{b}\right)\right)$.*

The proof can be found in Appendix C. And for the LIF model, define the weighted average firing rate $\hat{\mathbf{a}}[t] = \frac{\sum_{\tau=1}^{t}\lambda^{t-\tau}\mathbf{s}[\tau]}{\sum_{\tau=1}^{t}\lambda^{t-\tau}}$ and the weighted average inputs $\hat{\mathbf{x}}[t] = \frac{\sum_{\tau=0}^{t}\lambda^{t-\tau}\mathbf{x}[\tau]}{\sum_{\tau=0}^{t}\lambda^{t-\tau}}$, then we can similarly consider it as an approximation solver for the equilibrium state with random errors, as shown in Proposition 2. Please refer to Appendix E for details.

**Proposition 2.** *If the weighted average inputs converge to an equilibrium point $\hat{\mathbf{x}}[t] \to \mathbf{x}^*$, and there exists constant $c$ and $\gamma < 1$ such that $|\mathbf{u}_i^+[t]| \le c, \forall i, t$ and $\|\mathbf{W}\|_2 \le \gamma V_{th}$, then the weighted average firing rates $\hat{\mathbf{a}}[t]$ of FSNN with discrete LIF model gradually approximate an equilibrium point $\mathbf{a}^*$ with bounded random errors, which satisfies $\mathbf{a}^* = \sigma\left(\frac{1}{V_{th}}\left(\mathbf{W}\mathbf{a}^* + \mathbf{F}\mathbf{x}^* + \mathbf{b}\right)\right)$.*

## 4.2 Training of Feedback Spiking Neural Networks

Based on the derivation in Section 4.1, we can view the forward computation of FSNNs as a black-box solver for the fixed-point equilibrium equations, with some errors caused by finite time steps or the LIF model. Then we assume the (weighted) average firing rates $\mathbf{a}[T]$ after $T$ time steps approximately follow the equations. We will demonstrate how to train FSNNs and its biological plausibility.

### 4.2.1 Loss and Gradient Computation

Suppose that we simulate the SNN by $T$ time steps. Let $\mathbf{a}[T]$ denote the final (weighted) average firing rates. We configure a readout layer after these spiking neurons, which performs as a fully-connected classification layer, with the number of outputs as class numbers. We assume that these neurons will not spike or reset, and do classification based on the accumulated membrane potential. Then the outputs are equivalent as a linear transformation on $\mathbf{a}[T]$, i.e. $\mathbf{o} = \mathbf{W}^o\mathbf{a}[T]$. The loss $L$ is defined on $\mathbf{o}$ and labels $\mathbf{y}$ by commonly used loss functions $\mathcal{L}(\mathbf{o}, \mathbf{y})$, and we leverage the cross-entropy loss.

Let $f_\theta$ denote the function in fixed-point equation, e.g. $f_\theta(\mathbf{a}^*, \mathbf{x}^*) = \sigma\left(\frac{1}{V_{th}}\left(\mathbf{W}\mathbf{a}^* + \mathbf{F}\mathbf{x}^* + \mathbf{b}\right)\right)$. The gradient for parameters can be calculated based on the implicit differentiation as described in Section 3.2 by substituting $\mathbf{a}^*$ with $\mathbf{a}[T]$. Then parameters can be optimized based on common gradient descent methods, e.g. SGD [35] and its variants. A pseudocode is presented in Appendix B.

### 4.2.2 Biological Plausibility of Implicit Differentiation

While implicit differentiation may seem too abstract for information propagation compared with back-propagation, we will briefly discuss the biological possibility for this calculation and its connection to the Hebbian learning rule [10]. Consider the equilibrium state $\mathbf{a}^*$ following $\mathbf{a}^* = f_\theta(\mathbf{a}^*, \mathbf{x}^*) = \text{ReLU}\left(\frac{1}{V_{th}}\left(\mathbf{W}\mathbf{a}^* + \mathbf{F}\mathbf{x}^* + \mathbf{b}\right)\right)$. Let $\mathbf{m} = f_\theta'(\mathbf{a}^*, \mathbf{x}^*) = H\left(\frac{1}{V_{th}}\left(\mathbf{W}\mathbf{a}^* + \mathbf{F}\mathbf{x}^* + \mathbf{b}\right)\right) = H(\mathbf{a}^*), \mathbf{M} = \text{Diag}(\mathbf{m}), \widetilde{\mathbf{W}} = \mathbf{M}\mathbf{W}$, where $H(x) = \begin{cases} 1, x > 0 \\ 0, x \le 0 \end{cases}$. As indicated in Section 3.2, the gradient can be calculated as $\nabla_\theta\mathcal{L} = \left(\frac{\partial\mathcal{L}}{\partial\theta}\right)^T = \left(\frac{\partial f_\theta(\mathbf{a}^*, \mathbf{x}^*)}{\partial\theta}\right)^T\left(I - \frac{1}{V_{th}}\widetilde{\mathbf{W}}^T\right)^{-1}\left(\frac{\partial\mathcal{L}}{\partial\mathbf{a}^*}\right)^T$, and we can leverage a fixed-point update scheme to solve for $\beta^* = \left(I - \frac{1}{V_{th}}\widetilde{\mathbf{W}}^T\right)^{-1}\left(\frac{\partial\mathcal{L}}{\partial\mathbf{a}^*}\right)^T$ by iterating $\beta^{k+1} = \frac{1}{V_{th}}\widetilde{\mathbf{W}}^T\beta^k + \frac{\partial\mathcal{L}}{\partial\mathbf{a}^*}, \beta^k \to \beta^*$. This can be viewed as computing another equilibrium for these neurons: in this stage, neurons receive the inputs $\frac{\partial\mathcal{L}}{\partial\mathbf{a}^*}$ and they use the inverse directions of connections with a mask ($\mathbf{M}$ can be viewed as a mask matrix based on the firing condition in the first stage, which may be realized by some inhibition mechanisms) to compute for the equilibrium $\beta^*$. If $\mathbf{W}$ is symmetric and $V_{th} = 1$, it is similar to the energy-based method equilibrium propagation [36] to use the same weight as the forward computation for a second equilibrium computation. Plugging $\beta^*$ into the gradient and calculating $\left(\frac{\partial f_\theta(\mathbf{a}^*, \mathbf{x}^*)}{\partial\theta}\right)^T$ explicitly, we have: $\nabla_{\mathbf{W}}\mathcal{L} = \frac{1}{V_{th}}\mathbf{M}\beta^*\mathbf{a}^{*T}, \nabla_{\mathbf{F}}\mathcal{L} = \frac{1}{V_{th}}\mathbf{M}\beta^*\mathbf{x}^{*T}$. It is interesting to find that the change of weight

from neuron $j$ to neuron $i$ is proportional to the the equilibrium state of neuron $j$ in the first stage and that of neuron $i$ in the second stage, and is related to whether neuron $i$ fires in the first stage, because $\nabla_{\mathbf{W}_{i,j}} \mathcal{L} = \frac{1}{V_{th}} m_i \beta_i^* a_j^*$, $\nabla_{\mathbf{F}_{i,j}} \mathcal{L} = \frac{1}{V_{th}} m_i \beta_i^* x_j^*$. It is to some extent similar to the locally updated Hebbian learning rule $\Delta w_{i,j} \propto x_i x_j$ meaning that neurons wire together if they fire together [10], except that we take average firing rates and some temporal information (two stages) into account. Therefore, implicit differentiation calculation is only related to two equilibrium states by the neurons with a mask possibly realized by inhibition mechanisms, and may correspond to modified locally updated rules, which is more biologically plausible than BPTT with surrogate derivatives for SNNs. Please note that 'biological plausibility' here is a brief discussion in the context of the above properties, while there may be other aspects of biological implausibility as well.

## 4.3  Incorporating Multi-layer Structure into The Feedback Model

The multi-layer structure is commonly adopted in ANNs due to its stronger non-linearity and representation ability. To enhance the non-linearity of the fixed-point equilibrium equation, we propose to incorporate multi-layer structure into FSNNs as well. We configure $N$ subgroups of neurons as different layers, where the inputs have connections to the first layer, the $(l-1)$-th layer has connections to the $l$-th layer, and the last layer has feedback connections to the first layer. Let $\mathbf{u}^l(t)$ and $\mathbf{s}^l(t)$ denote the $l$-th layer, $\mathbf{x}(t)$ denote the inputs, $\mathbf{W}^1$ denote the feedback connection from the last layer to the first layer, and $\mathbf{F}^l$ denote the weight from the $(l-1)$-th layer (or input) to the $l$-th layer. The discrete update equations of membrane potentials are expressed as:

$$\begin{cases} \mathbf{u}^1[t+1] = \lambda \mathbf{u}^1[t] + \mathbf{W}^1 \mathbf{s}^N[t] + \mathbf{F}^1 \mathbf{x}[t] + \mathbf{b}^1 - V_{th} \mathbf{s}^1[t+1], \\ \mathbf{u}^{l+1}[t+1] = \lambda \mathbf{u}^{l+1}[t] + \mathbf{F}^{l+1} \mathbf{s}^l[t+1] + \mathbf{b}^{l+1} - V_{th} \mathbf{s}^{l+1}[t+1], \quad l = 1, 2, \cdots, N-1. \end{cases} \quad (10)$$

An illustration figure is presented in Appendix A. With similar definitions of average firing rates $\mathbf{a}^l[t]$ for different layers, and $\mathbf{u}_i^l[t] = \mathbf{u}_i^{l-}[t] + \mathbf{u}_i^{l+}[t]$, we have the equilibrium state as the following.

**Theorem 3.** *If the average inputs converge to an equilibrium point $\overline{\mathbf{x}}[t] \to \mathbf{x}^*$, and there exists constant $c$ and $\gamma < 1$ such that $|\mathbf{u}_i^{l+}[t]| \leq c, \forall i, l, t$ and $\|\mathbf{W}^1\|_2 \|\mathbf{F}^N\|_2 \cdots \|\mathbf{F}^2\|_2 \leq \gamma V_{th}^N$, then the average firing rates of multi-layer FSNN with discrete IF model will converge to equilibrium points $\mathbf{a}^l[t] \to \mathbf{a}^{l^*}$, which satisfy the fixed-point equations $\mathbf{a}^{1^*} = f_1\left(f_N \circ \cdots \circ f_2(\mathbf{a}^{1^*}), \mathbf{x}^*\right)$ and $\mathbf{a}^{l+1^*} = f_{l+1}(\mathbf{a}^{l^*})$, where $f_1(\mathbf{a}, \mathbf{x}) = \sigma\left(\frac{1}{V_{th}}(\mathbf{W}^1 \mathbf{a} + \mathbf{F}^1 \mathbf{x} + \mathbf{b}^1)\right)$ and $f_l(\mathbf{a}) = \sigma\left(\frac{1}{V_{th}}(\mathbf{F}^l \mathbf{a} + \mathbf{b}^l)\right)$.*

The proof can be found in Appendix D. There is also a similar proposition for the LIF model, please refer to Appendix E for details. We will do classification based on the (weighted) average firing rate of the last layer and calculate the implicit differentiation for the equation on $\mathbf{a}^N[T]$. The loss, solution for implicit differentiation, and optimization methods are the same as those in Section 4.2.1.

## 5  Experiments

In this section, we conduct extensive experiments to demonstrate the superior performance of our proposed method. Please refer to Appendix F for implementation details, including restrictions on the spectral norm and batch normalization, as well as training parameters. Since few previous methods leverage feedback architectures, we compare the results of our IDE method for FSNNs with most feedforward SNNs and report network structures[2] as well as the number of neurons and parameters during computation (calculated according to the papers or released codes) for comparison.

### 5.1  MNIST and Fashion-MNIST

We first evaluate our method on simple static image datasets including MNIST [19] and Fashion-MNIST [43], and compare the results with other directly trained SNNs [21, 41, 38, 14, 45, 46] or similar ANNs. Inputs are the same images with binary or real values at all time steps, which can be

---

[2]The notations are: '64C5' means a convolution with 64 output channels and kernel size 5, 's' after '64C5' means convolution with stride 2 while 'u' after that means a transposed convolution to upscale $2\times$, 'P2' means average pooling with size 2, '400' means fully-connected to 400 neurons, and 'F' means feedback layers.

regarded as input currents [46]. We leverage single-layer FSNNs, and adopt convolutional layers for MNIST while using fully-connected layers for Fashion-MNIST following [45]. As shown in Table 1, our models achieve comparable or better results with fewer neurons and parameters in a relatively small number of time steps, compared with other direct SNN training methods on feedforward or feedback architectures. Especially, our model achieves superior results on Fashion-MNIST with the similar structure in only 5 time steps. The LIF model performs slightly better than the IF model, probably because it leverages temporal information by encoding weighted average firing rates.

Table 1: Performance on MNIST and Fashion-MNIST. Results are based on 5 runs of experiments.

**MNIST**

| Method | Network structure | Time steps | Mean±Std | Best | Neurons | Params |
|---|---|---|---|---|---|---|
| BP [21] | 20C5-P2-50C5-P2-200 | >200 | / | 99.31% | 33K | 518K |
| STBP [41] | 15C5-P2-40C5-P2-300 | 30 | / | 99.42% | 26K | 607K |
| SLAYER [38] | 12C5-P2-64C5-P2 | 300 | 99.36%±0.05% | 99.41% | 28K | 51K |
| HM2BP [14] | 15C5-P2-40C5-P2-300 | 400 | 99.42%±0.11% | 99.49% | 26K | 607K |
| ST-RSBP [45] | 15C5-P2-40C5-P2-300 | 400 | **99.57%±0.04%** | **99.62%** | 26K | 607K |
| TSSL-BP [46] | 15C5-P2-40C5-P2-300 | 5 | 99.50%±0.02% | 99.53% | 26K | 607K |
| **IDE-IF (ours)** | 64C5s (F64C5) | 30 | 99.49%±0.04% | 99.55% | 13K | 229K |
| **IDE-LIF (ours)** | 64C5s (F64C5) | 30 | 99.53%±0.04% | 99.59% | 13K | 229K |

**Fashion-MNIST**

| Method | Network structure | Time steps | Mean±Std | Best | Neurons | Params |
|---|---|---|---|---|---|---|
| ANN [45] | 512-512 | / | / | 89.01% | 1.8K | 670K |
| HM2BP [45] | 400-400 | 400 | / | 88.99% | 1.6K | 478K |
| TSSL-BP [46] | 400-400 | 5 | 89.75%±0.03% | 89.80% | 1.6K | 478K |
| ST-RSBP [45] | 400 (F400) | 400 | 90.00%±0.14% | 90.13% | 1.2K | 478K |
| **IDE-IF (ours)** | 400 (F400) | **5** | **90.04%±0.09%** | **90.14%** | 1.2K | 478K |
| **IDE-LIF (ours)** | 400 (F400) | **5** | **90.07%±0.10%** | **90.25%** | 1.2K | 478K |

## 5.2 N-MNIST

We also evaluate our method on the neuromorphic dataset N-MNIST [28], whose inputs are spikes collected by dynamic vision sensors. We follow the same data pre-possessing as [46] and take 30 time steps, and we can view the (weighted) average inputs gradually converge to that at the last time step. Table 2 demonstrates the comparison results of our models and other directly trained models [14, 38, 46, 42]. It shows that our method can achieve satisfactory performance on neuromorphic data as well. Especially, only 30 time steps are required by our method for satisfactory performance.

Table 2: Performance on N-MNIST. Results are based on 5 runs of experiments.

| Method | Network structure | Time steps | Mean±Std | Best | Neurons | Params |
|---|---|---|---|---|---|---|
| HM2BP [14] | 400-400 | 600 | 98.88%±0.02% | 98.88% | 3K | 1.1M |
| SLAYER [38] | 500-500 | 300 | 98.89%±0.06% | 98.95% | 3K | 1.4M |
| SLAYER [38] | 12C5-P2-64C5-P2 | 300 | 99.20%±0.02% | 99.22% | 40K | 61K |
| TSSL-BP [46] | 12C5-P2-64C5-P2 | 30 | 99.23%±0.05% | 99.28% | 40K | 61K |
| STBP w/o NeuNorm [42] | CNN[1] | 60 | / | 99.44% | 414K | 17.3M |
| **IDE-IF (ours)** | 64C5s (F64C5) | 30 | 99.30%±0.04% | 99.35% | 21K | 291K |
| **IDE-LIF (ours)** | 64C5s (F64C5) | **30** | **99.42%±0.04%** | **99.47%** | 21K | 291K |

[1] 128C3-128C3-P2-128C3-256C3-P2-1024

## 5.3 CIFAR-10 and CIFAR-100

Then we evaluate our method on more complex CIFAR-10 and CIFAR-100 datasets [17]. We leverage multi-layer FSNNs with structures modified from AlexNet and CIFARNet proposed in [42], as indicated in the footnote of Table 3. We compare our model with SNNs converted from ANNs [37, 8, 32, 44] and directly trained SNNs [42, 46, 20, 40]. For CIFAR-100, no result of

directly trained SNN is reported, so we only compare with the converted ones. Table 3 demonstrates the superior results of our directly trained models with fewer neurons and parameters in a small number of time steps. Especially, our model can outperform the state-of-the-art SNN performance on CIFAR-100 with only 30 time steps, and achieves 1.59% accuracy improvement when 100 time steps are adopted. Please refer to Appendix G for more comparison results between IF and LIF models.

Table 3: Performance on CIFAR-10 and CIFAR-100. Results are based on 5 runs of experiments.

**CIFAR-10**

| Method | Network structure | Time steps | Mean±Std | Best | Neurons | Params |
|---|---|---|---|---|---|---|
| ANN-SNN [8] | CIFARNet | 400-600 | / | 90.61% | 726K | 45M |
| ANN-SNN [37] | VGG-16 | 2500 | / | 91.55% | 311K | 15M |
| ANN-SNN [8] | VGG-16 | 400-600 | / | 92.26% | 318K | 40M |
| Hybrid Training [32] | VGG-16 | 100 | / | 91.13% | 318K | 40M |
| STBP [42] | AlexNet | 12 | / | 85.24% | 595K | 21M |
| TSSL-BP [46] | AlexNet | 5 | 88.98%±0.27% | 89.22% | 595K | 21M |
| STBP [42] | CIFARNet | 12 | / | 90.53% | 726K | 45M |
| TSSL-BP [46] | CIFARNet | 5 | / | 91.41% | 726K | 45M |
| Surrogate gradient [20] | VGG-9 | 100 | / | 90.45% | 274K | 5.9M |
| ASF-BP [40] | VGG-7 | 400 | / | 91.35% | >240K | >30M |
| **IDE-LIF (ours)** | AlexNet-F | 30 | 91.74%±0.09% | 91.92% | 159K | 3.7M |
| **IDE-LIF (ours)** | AlexNet-F | 100 | **92.03%±0.07%** | **92.15%** | **159K** | **3.7M** |
| **IDE-LIF (ours)** | CIFARNet-F | 30 | 92.08%±0.14% | 92.23% | 232K | 11.8M |
| **IDE-LIF (ours)** | CIFARNet-F | 100 | **92.52%±0.17%** | **92.82%** | **232K** | **11.8M** |

[1] AlexNet [42]: 96C3-256C3-P2-384C3-P2-384C3-256C3-1024-1024
[2] AlexNet-F: 96C3s-256C3-384C3s-384C3-256C3 (F96C3u)
[3] CIFARNet [42]: 128C3-256C3-P2-512C3-P2-1024C3-512C3-1024-512
[4] CIFARNet-F: 128C3s-256C3-512C3s-1024C3-512C3 (F128C3u)

**CIFAR-100**

| Method | Network structure | Time steps | Mean±Std | Best | Neurons | Params |
|---|---|---|---|---|---|---|
| ANN [37] | VGG-16 | / | / | 71.22% | 311K | 15M |
| ANN-SNN [37] | VGG-16 | 2500 | / | 70.77% | 311K | 15M |
| ANN-SNN [8] | VGG-16 | 400-600 | / | 70.55% | 318K | 40M |
| ANN-SNN [44] | VGG-* | 300 | / | 71.84% | 540K | 9.7M |
| **IDE-IF (ours)** | CIFARNet-F | **30** | **71.56%±0.31%** | **72.10%** | 232K | 14.8M |
| **IDE-IF (ours)** | AlexNet-F | **100** | **72.02%±0.16%** | **72.23%** | **159K** | **5.2M** |
| **IDE-IF (ours)** | CIFARNet-F | **100** | **73.07%±0.21%** | **73.43%** | 232K | 14.8M |

## 5.4 Convergence to the Equilibrium

To verify the convergence of FSNNs to equilibrium states, we plot the difference norm on the fixed-point equation at each time step, i.e. $\|f_\theta(\mathbf{a}[t]) - \mathbf{a}[t]\|$, where $x = f_\theta(x)$ is the fixed-point equation and $\mathbf{a}[t]$ is the (weighted) average firing rate at time step $t$. Figure 1 demonstrates the convergence of different models. It is almost the same among different samples. Since the numerical precision of firing rates is only $\frac{1}{t}$, there would be a certain convergence error due to the finite time steps. And for the LIF model, there could be random errors compared with the IF model, as indicated in Section 4. The results conform to the theorems as the difference norm gradually decreases, i.e. firing rates converge to the equilibrium following the equation. It also indicates that exact precision is not necessary for satisfactory performance. Networks with fewer neurons converge faster, so a smaller number of time steps is needed, explaining why only 5 time steps are enough in the Fashion-MNIST experiment. For results on more datasets and different time steps, please refer to Appendix G.

## 5.5 Training Memory Costs

As described in the Introduction, an important advantage of our method is that we can avoid the large memory costs, from which the methods that backpropagate along the computational graph would suffer. To quantify this ease of training, we compare the GPU memory costs of our method and the

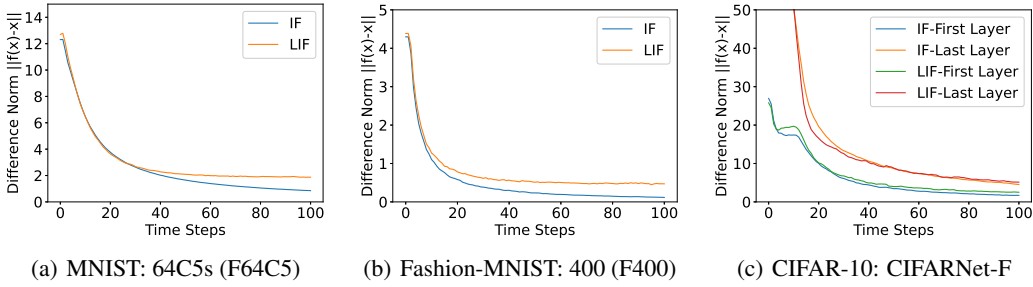

| (a) MNIST: 64C5s (F64C5) | (b) Fashion-MNIST: 400 (F400) | (c) CIFAR-10: CIFARNet-F |

Figure 1: Convergence to the equilibrium of different models on a random sample in 100 time steps.

representative STBP method [41, 42]. The architecture and training settings are the same, and the results are shown in Table 4. It well illustrates the smaller memory costs of our method, which is also agnostic to time steps. Meanwhile, our method could achieve higher performance.

## 5.6 Firing Sparsity

As for the efficient neuromorphic computation, the firing rate is an important statistic since the energy consumption is proportional to the number of spikes. We calculate the average firing rate of trained models, and compare the IF and LIF model trained by our IDE method, as well as the LIF model trained by STBP method [41, 42] with the same structure. The results in Table 5 demonstrate the firing sparsity of our model, as the average firing rate is only around or less than 0.7%. And it shows that the LIF model has a slightly sparser response compared with the IF model. We note that TSSL-BP [46] also reported the statistics about firing rate. According to their results, their trained model on CIFAR-10 has a roughly total 9.86% firing rate within 5 time steps. So it is interesting to find that our models have fewer spikes than theirs, even if we have more time steps (30 vs. 5), not to mention that our models have fewer neurons. The results also show that the model trained by our method has sparser spikes compared with STBP, demonstrating the superiority of our method.

Table 4: Comparison of training memory costs and accuracy between training methods. The model is trained on CIFAR-10 with AlexNet-F structure and LIF model.

| Method | Time steps | Accuracy | GPU memory |
|---|---|---|---|
| IDE (ours) | 30 | 91.74%±0.09% | 2.8G |
| STBP[*] | 30 | 87.18% | 11G |
| IDE (ours) | 100 | 92.03%±0.07% | 2.8G |
| STBP[*] | 100 | / | out of memory (≈36G) |

[*] Our implementation

Table 5: The average firing rate of trained models. The model is trained on CIFAR-10 with AlexNet-F structure and 30 time steps.

| Layer | IDE-IF | IDE-LIF | STBP-LIF[*] |
|---|---|---|---|
| Layer 1 | 0.0172 | 0.0166 | 0.0190 |
| Layer 2 | 0.0041 | 0.0039 | 0.0082 |
| Layer 3 | 0.0025 | 0.0024 | 0.0113 |
| Layer 4 | 0.0008 | 0.0008 | 0.0055 |
| Layer 5 | 0.0200 | 0.0177 | 0.0108 |
| Total | 0.0070 | **0.0066** | 0.0102 |

[*] Our implementation

## 6 Conclusion

In this work, we propose a novel training method for feedback spiking neural networks based on implicit differentiation on the equilibrium state. We first derive the equilibrium states of (weighted) average firing rates for the IF and the LIF models of FSNNs under both continuous and discrete views. Then we propose to optimize parameters of FSNNs only based on the implicit differentiation on the underlying fixed-point equation. This enables the backward procedure to be decoupled from the forward computational graph and therefore avoids the common training problems for SNNs, such as non-differentiability and large memory costs. Meanwhile, we briefly discuss the biological plausibility for the calculation of implicit differentiation, which only requires computing another equilibrium and is related to the locally updated Hebbian learning rule. Extensive experiments demonstrate the superior results of our method and models with fewer neurons and parameters in a small number of time steps, and the spikes are sparser in our trained models as well.

## Acknowledgement

Z. Lin was supported by the NSF China (No.s 61625301 and 61731018), NSFC Tianyuan Fund for Mathematics (No. 12026606) and Project 2020BD006 supported by PKU-Baidu Fund. Yisen Wang is partially supported by the National Natural Science Foundation of China under Grant 62006153, and Project 2020BD006 supported by PKU-Baidu Fund.

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
