# Supplementary Materials for: Training Feedback Spiking Neural Networks by Implicit Differentiation on the Equilibrium State

**Mingqing Xiao**[1], **Qingyan Meng**[2,3], **Zongpeng Zhang**[4], **Yisen Wang**[1,5], **Zhouchen Lin**[1,5,6*]

[1]Key Laboratory of Machine Perception (MoE), School of AI, Peking University
[2]The Chinese University of Hong Kong, Shenzhen
[3]Shenzhen Research Institute of Big Data
[4]Center for Data Science, Academy for Advanced Interdisciplinary Studies, Peking University
[5]Institute for Artificial Intelligence, Peking University
[6]Pazhou Lab, Guangzhou 510330, China
{mingqing_xiao, yisen.wang, zlin}@pku.edu.cn, qingyanmeng@link.cuhk.edu.cn,
zongpeng.zhang98@gmail.com

## A   Illustration Figure of Network Structures

Figure 1 illustrates our feedback models with single-layer and multi-layer structure as indicated in Sections 4.1 and 4.3.

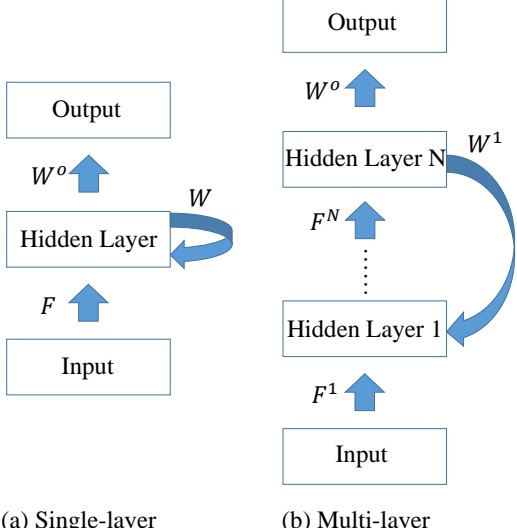

(a) Single-layer          (b) Multi-layer

Figure 1: Illustration of feedback models with single-layer and multi-layer structure.

## B   Pseudocode for the IDE Training Algorithm

We present the pseudocode of one iteration of IDE training in Algorithm 1 to better illustrate our training method.

---

*Corresponding author.

35th Conference on Neural Information Processing Systems (NeurIPS 2021).

**Algorithm 1** One iteration of IDE training.

---

**Input:** Network parameters $\theta$; Input data $x$; Label $y$; Time steps $T$; Other hyperparameters;
**Output:** Trained network parameters $\theta$.
  **Forward:**
 1: Simulate the SNN by T time steps with input $x$ based on Eq. (2) and calculate the final (weighted) average firing rate $a[T]$;
 2: Calculate the output $o$ and the loss $L$ based on $o$ and $y$.
  **Backward:**
 1: Specify the fixed-point equation $a = f_\theta(a)$ of the equilibrium state (define $g_\theta(a) = f_\theta(a) - a$);
 2: Calculate the gradients based on implicit differentiation:
 3:    (1) Solve the equation $\left(J_{g_\theta}^T|_{a[T]}\right)x^T + \left(\frac{\partial\mathcal{L}}{\partial a[T]}\right)^T = 0$ by root-finding methods;
 4:    (2) Calculate gradients $\frac{\partial\mathcal{L}}{\partial\theta} = -\frac{\partial\mathcal{L}}{\partial a[T]}\left(J_{g_\theta}^{-1}|_{a[T]}\right)\frac{\partial f_\theta(a[T])}{\partial\theta}$ based on the solution and $\frac{\partial f_\theta(a[T])}{\partial\theta}$;
 5: Update $\theta$ based on the gradient-based optimizer.

---

## C  Proof of Theorem 1 and Theorem 2

We first prove Theorem 1. Then Theorem 2 is similarly proved.

**Theorem 1.** *If the average inputs converge to an equilibrium point $\overline{\mathbf{x}}(t) \to \mathbf{x}^*$, and there exists constant $c$ and $\gamma < 1$ such that $|\mathbf{u}_i^+[t]| \le c, \forall i, t$ and $\|\mathbf{W}\|_2 \le \gamma V_{th}$, then the average firing rates of FSNN with continuous IF model will converge to an equilibrium point $\mathbf{a}(t) \to \mathbf{a}^*$, which satisfies the fixed-point equation $\mathbf{a}^* = ReLU\left(\frac{1}{V_{th}}\left(\mathbf{W}\mathbf{a}^* + \mathbf{F}\mathbf{x}^* + \mathbf{b}\right)\right)$.*

*Proof.* $\forall 0 \le \epsilon_t < \Delta t_d$, we construct the sequence $\{\mathbf{a}_{\epsilon_t}^i\}_{i=0}^\infty$ where $\mathbf{a}_{\epsilon_t}^i = \mathbf{a}\left(t_{\epsilon_t}^i\right), t_{\epsilon_t}^i = i\Delta t_d + \epsilon_t$. Then the equation (1) turns into the iterative equation (2) as the following:

$$\mathbf{a}(t) = \text{ReLU}\left(\frac{1}{V_{th}}\left(\frac{t-\Delta t_d}{t}\mathbf{W}\mathbf{a}(t-\Delta t_d) + \mathbf{F}\overline{\mathbf{x}}(t) + \mathbf{b}\right)\right) - \frac{1}{V_{th}}\frac{\mathbf{u}^+(t)}{t}, \tag{1}$$

$$\mathbf{a}_{\epsilon_t}^{i+1} = \text{ReLU}\left(\frac{1}{V_{th}}\left(\frac{t_{\epsilon_t}^{i+1}-\Delta t_d}{t_{\epsilon_t}^i}\mathbf{W}\mathbf{a}_{\epsilon_t}^i + \mathbf{F}\overline{\mathbf{x}}(t_{\epsilon_t}^{i+1}) + \mathbf{b}\right)\right) - \frac{1}{V_{th}}\frac{\mathbf{u}^+(t_{\epsilon_t}^{i+1})}{t_{\epsilon_t}^{i+1}}. \tag{2}$$

We prove the sequence $\{\mathbf{a}_{\epsilon_t}^i\}_{i=0}^\infty$ converges. Consider $\|\mathbf{a}_{\epsilon_t}^{i+1} - \mathbf{a}_{\epsilon_t}^i\|$, it satisfies:

$$\begin{aligned}
&\left\|\mathbf{a}_{\epsilon_t}^{i+1} - \mathbf{a}_{\epsilon_t}^i\right\| \\
=\ &\left\|\left(\text{ReLU}\left(\frac{1}{V_{th}}\left(\frac{t_{\epsilon_t}^{i+1}-\Delta t_d}{t_{\epsilon_t}^{i+1}}\mathbf{W}\mathbf{a}_{\epsilon_t}^i + \mathbf{F}\overline{\mathbf{x}}(t_{\epsilon_t}^{i+1}) + \mathbf{b}\right)\right) - \frac{1}{V_{th}}\frac{\mathbf{u}^+(t_{\epsilon_t}^{i+1})}{t_{\epsilon_t}^{i+1}}\right) \right.\\
&\left. - \left(\text{ReLU}\left(\frac{1}{V_{th}}\left(\frac{t_{\epsilon_t}^i-\Delta t_d}{t_{\epsilon_t}^i}\mathbf{W}\mathbf{a}_{\epsilon_t}^{i-1} + \mathbf{F}\overline{\mathbf{x}}(t_{\epsilon_t}^i) + \mathbf{b}\right)\right) - \frac{1}{V_{th}}\frac{\mathbf{u}^+(t_{\epsilon_t}^i)}{t_{\epsilon_t}^i}\right)\right\| \\
\le\ &\left\|\text{ReLU}\left(\frac{1}{V_{th}}\left(\mathbf{W}\mathbf{a}_{\epsilon_t}^i + \mathbf{F}\mathbf{x}^* + \mathbf{b}\right)\right) - \text{ReLU}\left(\frac{1}{V_{th}}\left(\mathbf{W}\mathbf{a}_{\epsilon_t}^{i-1} + \mathbf{F}\mathbf{x}^* + \mathbf{b}\right)\right)\right\| \\
&+ \left\|\text{ReLU}\left(\frac{1}{V_{th}}\left(\frac{t_{\epsilon_t}^{i+1}-\Delta t_d}{t_{\epsilon_t}^{i+1}}\mathbf{W}\mathbf{a}_{\epsilon_t}^i + \mathbf{F}\overline{\mathbf{x}}(t_{\epsilon_t}^{i+1}) + \mathbf{b}\right)\right) - \frac{1}{V_{th}}\frac{\mathbf{u}^+(t_{\epsilon_t}^{i+1})}{t_{\epsilon_t}^{i+1}} - \text{ReLU}\left(\frac{1}{V_{th}}\left(\mathbf{W}\mathbf{a}_{\epsilon_t}^i + \mathbf{F}\mathbf{x}^* + \mathbf{b}\right)\right)\right\| \\
&+ \left\|\text{ReLU}\left(\frac{1}{V_{th}}\left(\frac{t_{\epsilon_t}^i-\Delta t_d}{t_{\epsilon_t}^i}\mathbf{W}\mathbf{a}_{\epsilon_t}^{i-1} + \mathbf{F}\overline{\mathbf{x}}(t_{\epsilon_t}^i) + \mathbf{b}\right)\right) - \frac{1}{V_{th}}\frac{\mathbf{u}^+(t_{\epsilon_t}^i)}{t_{\epsilon_t}^i} - \text{ReLU}\left(\frac{1}{V_{th}}\left(\mathbf{W}\mathbf{a}_{\epsilon_t}^{i-1} + \mathbf{F}\mathbf{x}^* + \mathbf{b}\right)\right)\right\| \\
\le\ &\left\|\text{ReLU}\left(\frac{1}{V_{th}}\left(\mathbf{W}\mathbf{a}_{\epsilon_t}^i + \mathbf{F}\mathbf{x}^* + \mathbf{b}\right)\right) - \text{ReLU}\left(\frac{1}{V_{th}}\left(\mathbf{W}\mathbf{a}_{\epsilon_t}^{i-1} + \mathbf{F}\mathbf{x}^* + \mathbf{b}\right)\right)\right\| \\
&+ \frac{1}{V_{th}}\left(\left\|\frac{\Delta t_d}{t_{\epsilon_t}^{i+1}}\mathbf{W}\mathbf{a}_{\epsilon_t}^i\right\| + \left\|\mathbf{F}\left(\overline{\mathbf{x}}(t_{\epsilon_t}^{i+1}) - \mathbf{x}^*\right)\right\| + \left\|\frac{\mathbf{u}^+(t_{\epsilon_t}^{i+1})}{t_{\epsilon_t}^{i+1}}\right\| + \left\|\frac{\Delta t_d}{t_{\epsilon_t}^i}\mathbf{W}\mathbf{a}_{\epsilon_t}^{i-1}\right\| + \left\|\mathbf{F}\left(\overline{\mathbf{x}}(t_{\epsilon_t}^i) - \mathbf{x}^*\right)\right\| + \left\|\frac{\mathbf{u}^+(t_{\epsilon_t}^i)}{t_{\epsilon_t}^i}\right\|\right).
\end{aligned} \tag{3}$$

As $\|\mathbf{W}\|_2 \leq \gamma V_{th}, \gamma < 1, \bar{\mathbf{x}}(t) \to \mathbf{x}^*$, and $|\mathbf{u}_i^+(t)| \leq c, \forall i, t$, we have $\|\mathbf{a}_{\epsilon_t}^{i+1}\| \leq \gamma \|\mathbf{a}_{\epsilon_t}^i\| + \frac{1}{V_{th}} \left( \|\mathbf{F}\bar{\mathbf{x}}(t_{\epsilon_t}^{i+1})\| + \|\mathbf{b}\| + \left\| \frac{\mathbf{u}^+(t_{\epsilon_t}^{i+1})}{t_{\epsilon_t}^{i+1}} \right\| \right)$, and therefore $\|\mathbf{a}_{\epsilon_t}^i\|$ is bounded.

Since $t_{\epsilon_t}^i \to \infty, \bar{\mathbf{x}}(t) \to \mathbf{x}^*$, and $|\mathbf{u}_i^+(t)| \leq c, \forall i, t$, then $\forall \epsilon > 0, \exists M$ such that when $i > M$, we have:

$$
\frac{1}{V_{th}} \left( \left\| \frac{\Delta t_d}{t_{\epsilon_t}^{i+1}} \mathbf{W} \mathbf{a}_{\epsilon_t}^i \right\| + \|\mathbf{F}\left(\bar{\mathbf{x}}(t_{\epsilon_t}^{i+1}) - \mathbf{x}^*\right)\| + \left\| \frac{\mathbf{u}^+(t_{\epsilon_t}^{i+1})}{t_{\epsilon_t}^{i+1}} \right\| + \left\| \frac{\Delta t_d}{t_{\epsilon_t}^i} \mathbf{W} \mathbf{a}_{\epsilon_t}^{i-1} \right\| + \|\mathbf{F}\left(\bar{\mathbf{x}}(t_{\epsilon_t}^i) - \mathbf{x}^*\right)\|
$$
$$
+ \left\| \frac{\mathbf{u}^+(t_{\epsilon_t}^i)}{t_{\epsilon_t}^i} \right\| \right) \leq \frac{\epsilon(1-\gamma)}{2}. \tag{4}
$$

And since $\|\mathbf{W}\|_2 \leq \gamma V_{th}$, we have:

$$
\left\| \text{ReLU}\left( \frac{1}{V_{th}} \left( \mathbf{W}\mathbf{a}_{\epsilon_t}^i + \mathbf{F}\mathbf{x}^* + \mathbf{b} \right) \right) - \text{ReLU}\left( \frac{1}{V_{th}} \left( \mathbf{W}\mathbf{a}_{\epsilon_t}^{i-1} + \mathbf{F}\mathbf{x}^* + \mathbf{b} \right) \right) \right\| \leq \gamma \left\| \mathbf{a}_{\epsilon_t}^i - \mathbf{a}_{\epsilon_t}^{i-1} \right\|. \tag{5}
$$

Therefore, when $i > M$ it holds that:

$$
\left\| \mathbf{a}_{\epsilon_t}^{i+1} - \mathbf{a}_{\epsilon_t}^i \right\| \leq \gamma \left\| \mathbf{a}_{\epsilon_t}^i - \mathbf{a}_{\epsilon_t}^{i-1} \right\| + \frac{\epsilon(1-\gamma)}{2}. \tag{6}
$$

By iterating the above inequality, we have $\|\mathbf{a}_{\epsilon_t}^{i+1} - \mathbf{a}_{\epsilon_t}^i\| \leq \gamma^{i-M}\|\mathbf{a}_{\epsilon_t}^{M+1} - \mathbf{a}_{\epsilon_t}^M\| + \frac{\epsilon(1-\gamma)}{2}\left(1 + \gamma + \cdots + \gamma^{i-M-1}\right) < \gamma^{i-M}\|\mathbf{a}_{\epsilon_t}^{M+1} - \mathbf{a}_{\epsilon_t}^M\| + \frac{\epsilon}{2}$. There exists $M'$ such that when $i > M + M'$, $\gamma^{i-M}\|\mathbf{a}_{\epsilon_t}^{M+1} - \mathbf{a}_{\epsilon_t}^M\| \leq \frac{\epsilon}{2}$, and therefore $\|\mathbf{a}_{\epsilon_t}^{i+1} - \mathbf{a}_{\epsilon_t}^i\| < \epsilon$. According to Cauchy's convergence test, the sequence $\{\mathbf{a}_{\epsilon_t}^i\}_{i=0}^\infty$ converges to $\mathbf{a}_{\epsilon_t}^*$. Considering the limit, it satisfies $\mathbf{a}_{\epsilon_t}^* = \text{ReLU}\left(\frac{1}{V_{th}}\left(\mathbf{W}\mathbf{a}_{\epsilon_t}^* + \mathbf{F}\mathbf{x}^* + \mathbf{b}\right)\right)$.

The solution of $\mathbf{a}$ for the equation $\mathbf{a} = \text{ReLU}\left(\frac{1}{V_{th}}\left(\mathbf{W}\mathbf{a} + \mathbf{F}\mathbf{x}^* + \mathbf{b}\right)\right)$ is unique, since $\|\mathbf{W}\|_2 \leq \gamma V_{th}, \gamma < 1$. So $\forall \epsilon_t$, the sequence $\{\mathbf{a}_{\epsilon_t}^i\}_{i=0}^\infty$ converges to the same point. Therefore, the average firing rates $\mathbf{a}(t)$ of IF model will converge to an equilibrium point $\mathbf{a}(t) \to \mathbf{a}^*$, which satisfies the fixed-point equation $\mathbf{a}^* = \text{ReLU}\left(\frac{1}{V_{th}}\left(\mathbf{W}\mathbf{a}^* + \mathbf{F}\mathbf{x}^* + \mathbf{b}\right)\right)$.

$\square$

Theorem 2 can be similarly proved as the above proof for sequence convergence, by substituting the ReLU function with $\sigma(x) = \begin{cases} 1, & x > 1 \\ x, & 0 \leq x \leq 1 \\ 0, & x < 0 \end{cases}$. We omit repetitive details here.

**Theorem 2.** *If the average inputs converge to an equilibrium point $\bar{\mathbf{x}}[t] \to \mathbf{x}^*$, and there exists constant $c$ and $\gamma < 1$ such that $|\mathbf{u}_i^+(t)| \leq c, \forall i, t$ and $\|\mathbf{W}\|_2 \leq \gamma V_{th}$, then the average firing rates of FSNN with discrete IF model will converge to an equilibrium point $\mathbf{a}[t] \to \mathbf{a}^*$, which satisfies the fixed-point equation $\mathbf{a}^* = \sigma\left(\frac{1}{V_{th}}\left(\mathbf{W}\mathbf{a}^* + \mathbf{F}\mathbf{x}^* + \mathbf{b}\right)\right)$.*

## D  Proof of Theorem 3

**Theorem 3.** *If the average inputs converge to an equilibrium point $\bar{\mathbf{x}}[t] \to \mathbf{x}^*$, and there exists constant $c$ and $\gamma < 1$ such that $|\mathbf{u}_i^{l+}[t]| \leq c, \forall i, l, t$ and $\|\mathbf{W}^1\|_2\|\mathbf{F}^N\|_2 \cdots \|\mathbf{F}^2\|_2 \leq \gamma V_{th}^N$, then the average firing rates of multi-layer FSNN with discrete IF model will converge to equilibrium points $\mathbf{a}^l[t] \to \mathbf{a}^{l*}$, which satisfy the fixed-point equations $\mathbf{a}^{1*} = f_1\left(f_N \circ \cdots \circ f_2(\mathbf{a}^{1*}), \mathbf{x}^*\right)$ and $\mathbf{a}^{l+1*} = f_{l+1}(\mathbf{a}^{l*})$, where $f_1(\mathbf{a}, \mathbf{x}) = \sigma\left(\frac{1}{V_{th}}(\mathbf{W}^1\mathbf{a} + \mathbf{F}^1\mathbf{x} + \mathbf{b}^1)\right), f_{l+1}(\mathbf{a}) = \sigma\left(\frac{1}{V_{th}}(\mathbf{F}^{l+1}\mathbf{a} + \mathbf{b}^{l+1})\right)$.*

*Proof.* When the multi-layer structure is considered, with similar definitions of average firing rates $\mathbf{a}^l[t]$ for different layers and the separation $\mathbf{u}_i^l[t] = \mathbf{u}_i^{l\,-}[t] + \mathbf{u}_i^{l\,+}[t]$, we have the equations:

$$
\begin{cases}
\mathbf{a}^1[t+1] = \sigma\left(\dfrac{1}{V_{th}}\left(\dfrac{t}{t+1}\mathbf{W}^1\mathbf{a}^N[t] + \mathbf{F}^1\overline{\mathbf{x}}[t] + \mathbf{b}^1\right)\right) - \dfrac{1}{V_{th}}\dfrac{\mathbf{u}^{1\,+}[t+1]}{t+1}, \\[2mm]
\mathbf{a}^{l+1}[t+1] = \sigma\left(\dfrac{1}{V_{th}}\left(\mathbf{F}^{l+1}\mathbf{a}^l[t+1] + \mathbf{b}^{l+1}\right)\right) - \dfrac{1}{V_{th}}\dfrac{\mathbf{u}^{l+1\,+}[t+1]}{t+1}, \quad l = 1, \cdots, N-1.
\end{cases}
$$
(7)

Let $f_1^{t+1}(\mathbf{a}, \mathbf{x}, \mathbf{u}^+) = \sigma\left(\frac{1}{V_{th}}\left(\frac{t}{t+1}\mathbf{W}^1\mathbf{a} + \mathbf{F}^1\mathbf{x} + \mathbf{b}^1\right)\right) - \frac{1}{V_{th}}\frac{\mathbf{u}^+}{t+1}$,

$f_{l+1}^t(\mathbf{a}, \mathbf{u}^+) = \sigma\left(\frac{1}{V_{th}}\left(\mathbf{F}^{l+1}\mathbf{a} + \mathbf{b}^{l+1}\right)\right) - \frac{1}{V_{th}}\frac{\mathbf{u}^+}{t}$,

$f_1(\mathbf{a}, \mathbf{x}) = \sigma\left(\frac{1}{V_{th}}(\mathbf{W}^1\mathbf{a} + \mathbf{F}^1\mathbf{x} + \mathbf{b}^1)\right)$, $f_{l+1}(\mathbf{a}) = \sigma\left(\frac{1}{V_{th}}(\mathbf{F}^{l+1}\mathbf{a} + \mathbf{b}^{l+1})\right)$.

Then $\mathbf{a}^1[t+1] = f_1^{t+1}\left(f_N^t\left(\cdots f_2^t\left(\mathbf{a}^1[t], \mathbf{u}^{2\,+}[t]\right)\cdots, \mathbf{u}^{N\,+}[t]\right), \overline{\mathbf{x}}[t], \mathbf{u}^{1\,+}[t+1]\right)$.

We have:

$$
\begin{aligned}
&\left\|\mathbf{a}^1[t+1] - \mathbf{a}^1[t]\right\| \\
=\ &\left\|f_1^{t+1}\left(f_N^t\left(\cdots f_2^t\left(\mathbf{a}^1[t], \mathbf{u}^{2\,+}[t]\right)\cdots, \mathbf{u}^{N\,+}[t]\right), \overline{\mathbf{x}}[t], \mathbf{u}^{1\,+}[t+1]\right)\right. \\
&\left. - f_1^t\left(f_N^{t-1}\left(\cdots f_2^{t-1}\left(\mathbf{a}^1[t-1], \mathbf{u}^{2\,+}[t-1]\right)\cdots, \mathbf{u}^{N\,+}[t-1]\right), \overline{\mathbf{x}}[t-1], \mathbf{u}^{1\,+}[t]\right)\right\| \\
\leq\ &\left\|f_1\left(f_N\left(\cdots f_2\left(\mathbf{a}^1[t]\right)\cdots\right), \mathbf{x}^*\right) - f_1\left(f_N\left(\cdots f_2\left(\mathbf{a}^1[t-1]\right)\cdots\right), \mathbf{x}^*\right)\right\| \\
&+ \left\|f_1^{t+1}\left(f_N^t\left(\cdots f_2^t\left(\mathbf{a}^1[t], \mathbf{u}^2 + [t]\right)\cdots, \mathbf{u}^{N\,+}[t]\right), \overline{\mathbf{x}}[t], \mathbf{u}^{1\,+}[t+1]\right) - f_1\left(f_N\left(\cdots f_2\left(\mathbf{a}^1[t]\right)\cdots\right), \mathbf{x}^*\right)\right\| \\
&+ \left\|f_1^t\left(f_N^{t-1}\left(\cdots f_2^{t-1}\left(\mathbf{a}^1[t-1], \mathbf{u}^{2\,+}[t-1]\right)\cdots, \mathbf{u}^{N\,+}[t-1]\right), \overline{\mathbf{x}}[t-1], \mathbf{u}^{1\,+}[t]\right)\right. \\
&\left. - f_1\left(f_N\left(\cdots f_2\left(\mathbf{a}^1[t-1]\right)\cdots\right), \mathbf{x}^*\right)\right\| \\
\leq\ &\left\|f_1\left(f_N\left(\cdots f_2\left(\mathbf{a}^1[t]\right)\cdots\right), \mathbf{x}^*\right) - f_1\left(f_N\left(\cdots f_2\left(\mathbf{a}^1[t-1]\right)\cdots\right), \mathbf{x}^*\right)\right\| \\
&+ \frac{1}{V_{th}}\left(\left\|\frac{1}{t+1}\mathbf{W}^1 f_N^t\left(\cdots f_2^t\left(\mathbf{a}^1[t], \mathbf{u}^{2\,+}[t]\right)\cdots, \mathbf{u}^{N\,+}[t]\right)\right\|\right. \\
&+ \underbrace{\left\|\mathbf{W}^1\left(f_N^t\left(\cdots f_2^t\left(\mathbf{a}^1[t], \mathbf{u}^{2\,+}[t]\right)\cdots, \mathbf{u}^{N\,+}[t]\right) - f_N\left(\cdots f_2\left(\mathbf{a}^1[t]\right)\cdots\right)\right)\right\|}_{A} \\
&+ \left\|\mathbf{F}\left(\overline{\mathbf{x}}[t] - \mathbf{x}^*\right)\right\| + \left\|\frac{\mathbf{u}^{1\,+}[t+1]}{t+1}\right\| \\
&+ \left\|\frac{1}{t}\mathbf{W}^1 f_N^{t-1}\left(\cdots f_2^{t-1}\left(\mathbf{a}^1[t-1], \mathbf{u}^{2\,+}[t-1]\right)\cdots, \mathbf{u}^{N\,+}[t-1]\right)\right\| \\
&+ \underbrace{\left\|\mathbf{W}^1\left(f_N^{t-1}\left(\cdots f_2^{t-1}\left(\mathbf{a}^1[t-1], \mathbf{u}^{2\,+}[t-1]\right)\cdots, \mathbf{u}^{N\,+}[t-1]\right) - f_N\left(\cdots f_2\left(\mathbf{a}^1[t-1]\right)\cdots\right)\right)\right\|}_{B} \\
&+ \left\|\mathbf{F}\left(\overline{\mathbf{x}}[t-1] - \mathbf{x}^*\right)\right\| + \left\|\frac{\mathbf{u}^{1\,+}[t]}{t}\right\|\bigg).
\end{aligned}
$$
(8)

For the term $A$ and $B$, they are bounded by:

$$A \leq \frac{1}{V_{th}} \left( \left\| \mathbf{W}^1 \mathbf{F}^N \left( f_{N-1}^t \left( \cdots f_2^t \left( \mathbf{a}^1[t], \mathbf{u}^{2^+}[t] \right) \cdots, \mathbf{u}^{N-1^+} \right) - f_{N-1} \left( \cdots f_2 \left( \mathbf{a}^1[t] \right) \cdots \right) \right) \right\| + \left\| \mathbf{W}^1 \frac{\mathbf{u}^{N^+}[t]}{t} \right\| \right)$$

$$\leq \quad \cdots \cdots$$

$$\leq \quad \frac{1}{V_{th}} \left\| \mathbf{W}^1 \frac{\mathbf{u}^{N^+}[t]}{t} \right\| + \cdots + \frac{1}{V_{th}^{N-1}} \left\| \mathbf{W}^1 \mathbf{F}^N \cdots \mathbf{F}^3 \frac{\mathbf{u}^{2^+}[t]}{t} \right\|,$$

$$(9)$$

and $B$ has the same form as $A$ by substituting $t$ with $t-1$.

Since $\|\mathbf{W}^1\|_2 \|\mathbf{F}^N\|_2 \cdots \|\mathbf{F}^2\|_2 \leq \gamma V_{th}^N$, we have:

$$\left\| f_1 \left( f_N \left( \cdots f_2 \left( \mathbf{a}^1[t] \right) \cdots \right), \mathbf{x}^* \right) - f_1 \left( f_N \left( \cdots f_2 \left( \mathbf{a}^1[t-1] \right) \cdots \right), \mathbf{x}^* \right) \right\|$$

$$\leq \quad \left\| \frac{1}{V_{th}} \mathbf{W}^1 \left( f_N \left( \cdots f_2 \left( \mathbf{a}^1[t] \right) \cdots \right) - f_N \left( \cdots f_2 \left( \mathbf{a}^1[t-1] \right) \cdots \right) \right) \right\|$$

$$\leq \quad \cdots \cdots \tag{10}$$

$$\leq \quad \left\| \frac{1}{V_{th}^N} \mathbf{W}^1 \mathbf{F}^N \cdots \mathbf{F}^2 \left( \mathbf{a}^1[t] - \mathbf{a}^1[t-1] \right) \right\|$$

$$\leq \quad \gamma \left\| \mathbf{a}^1[t] - \mathbf{a}^1[t-1] \right\|.$$

And since $t \to \infty, \bar{\mathbf{x}}[t] \to \mathbf{x}^*, |\mathbf{u}_i^{l^+}[t]| \leq c, \forall i, l, t$, then $\forall \epsilon > 0, \exists M$ such that when $t > M$, we have:

$$\left\| \mathbf{a}^1[t+1] - \mathbf{a}^1[t] \right\| \leq \gamma \left\| \mathbf{a}^1[t] - \mathbf{a}^1[t-1] \right\| + \frac{\epsilon(1-\gamma)}{2}. \tag{11}$$

Then $\|\mathbf{a}^1[t+1] - \mathbf{a}^1[t]\| < \gamma^{t-M} \|\mathbf{a}^1[M+1] - \mathbf{a}^1[M]\| + \frac{\epsilon}{2}$, and there exists $M'$ such that when $t > M + M'$, $\|\mathbf{a}^1[t+1] - \mathbf{a}^1[t]\| < \epsilon$. According to Cauchy's convergence test, $\mathbf{a}^1[t]$ converges to $\mathbf{a}^{1^*}$, which satisfies $\mathbf{a}^{1^*} = f_1 \left( f_N \circ \cdots \circ f_2(\mathbf{a}^{1^*}), \mathbf{x}^* \right)$. Considering the limit, $\mathbf{a}^{l+1}[t]$ converges to $\mathbf{a}^{l+1^*}$, which satisfies $\mathbf{a}^{l+1^*} = f_{l+1}(\mathbf{a}^{l^*})$.

$\square$

# E   Derivation for the LIF Model

## E.1   Continuous View

We follow the same notations as Section 4.1.1 and redefine $\mathbf{W}, \mathbf{F}, \mathbf{b}$ by absorbing $\tau_m$ into them. The dynamics of membrane potentials are expressed as:

$$\frac{d\mathbf{u}}{dt} = -\frac{1}{\tau_m} \mathbf{u} + \mathbf{W}\mathbf{s}(t - \Delta t_d) + \mathbf{F}\mathbf{x}(t) + \mathbf{b} - V_{th}\mathbf{s}(t). \tag{12}$$

Through integration, we have:

$$\mathbf{u}(t) = \mathbf{W} \int_0^{t-\Delta t_d} \kappa(t - \Delta t_d - \tau)\mathbf{s}(\tau)d\tau + \mathbf{F} \int_0^t \kappa(t - \tau)\mathbf{x}(\tau)d\tau + t\mathbf{b} - V_{th} \int_0^t \kappa(t - \tau)\mathbf{s}(\tau)d\tau, \tag{13}$$

where $\kappa(\tau) = \exp(-\frac{\tau}{\tau_m})$ is the response kernel of the LIF model. Define the weighted average firing rate as $\hat{\mathbf{a}}(t) = \frac{\int_0^t \kappa(t-\tau)\mathbf{s}(\tau)d\tau}{\int_0^t \kappa(t-\tau)d\tau}$, and the weighted average inputs as $\hat{\mathbf{x}}(t) = \frac{\int_0^t \kappa(t-\tau)\mathbf{x}(\tau)d\tau}{\int_0^t \kappa(t-\tau)d\tau}$. Then we have the equation:

$$\hat{\mathbf{a}}(t) = \frac{1}{V_{th}} \left( \frac{\int_0^{t-\Delta t_d} \kappa(\tau)d\tau}{\int_0^t \kappa(\tau)d\tau} \mathbf{W}\hat{\mathbf{a}}(t - \Delta t_d) + \mathbf{F}\hat{\mathbf{x}}(t) + \mathbf{b} - \frac{\mathbf{u}(t)}{\int_0^t \kappa(\tau)d\tau} \right). \tag{14}$$

Similarly, we can divide $\mathbf{u}(t)$ into two parts $\mathbf{u}_i(t) = \mathbf{u}_i^-(t) + \mathbf{u}_i^+(t)$, and we have the equation with the element-wise ReLU function and a bounded $\mathbf{u}^+(t)$:

$$\hat{\mathbf{a}}(t) = \text{ReLU}\left(\frac{1}{V_{th}}\left(\frac{\int_0^{t-\Delta t_d} \kappa(\tau)\mathrm{d}\tau}{\int_0^t \kappa(\tau)\mathrm{d}\tau}\mathbf{W}\hat{\mathbf{a}}(t-\Delta t_d) + \mathbf{F}\hat{\mathbf{x}}(t) + \mathbf{b}\right)\right) - \frac{1}{V_{th}}\frac{\mathbf{u}^+(t)}{\int_0^t \kappa(\tau)\mathrm{d}\tau}. \quad (15)$$

As $\int_0^t \kappa(\tau)\mathrm{d}\tau = \tau_m\left(1 - \exp(-\frac{t}{\tau_m})\right) \to \tau_m$, compared with the IF model, there could be random error caused by $\frac{\mathbf{u}^+(t)}{\int_0^t \kappa(\tau)\mathrm{d}\tau}$ which are not eliminated with time $t \to \infty$. Therefore when the weighted average inputs converge to an equilibrium point $\hat{\mathbf{x}}(t) \to \mathbf{x}^*$, the LIF model only gradually approximates an equilibrium with some random error, and the equilibrium state $\mathbf{a}^*$ still follows the equation $\mathbf{a}^* = \text{ReLU}\left(\frac{1}{V_{th}}(\mathbf{W}\mathbf{a}^* + \mathbf{F}\mathbf{x}^* + \mathbf{b})\right)$. The error is:

$$
\begin{aligned}
e(t) = \quad & \left\|\text{ReLU}\left(\frac{1}{V_{th}}(\mathbf{W}\hat{\mathbf{a}}(t) + \mathbf{F}\mathbf{x}^* + \mathbf{b})\right) - \hat{\mathbf{a}}(t)\right\| \\
= \quad & \left\|\text{ReLU}\left(\frac{1}{V_{th}}(\mathbf{W}\hat{\mathbf{a}}(t) + \mathbf{F}\mathbf{x}^* + \mathbf{b})\right) - \text{ReLU}\left(\frac{1}{V_{th}}\left(\frac{\int_0^{t-\Delta t_d} \kappa(\tau)\mathrm{d}\tau}{\int_0^t \kappa(\tau)\mathrm{d}\tau}\mathbf{W}\hat{\mathbf{a}}(t-\Delta t_d) + \mathbf{F}\hat{\mathbf{x}}(t) + \mathbf{b}\right)\right)\right. \\
& \left. + \frac{1}{V_{th}}\frac{\mathbf{u}^+(t)}{\int_0^t \kappa(\tau)\mathrm{d}\tau}\right\| \\
\leq \quad & \frac{1}{V_{th}}\left(\|\mathbf{W}(\hat{\mathbf{a}}(t) - \hat{\mathbf{a}}(t-\Delta t_d))\| + \left\|\frac{\int_{t-\Delta t_d}^t \kappa(\tau)\mathrm{d}\tau}{\int_0^t \kappa(\tau)\mathrm{d}\tau}\mathbf{W}\hat{\mathbf{a}}(t-\Delta t_d)\right\| + \|\mathbf{F}(\hat{\mathbf{x}}(t) - \mathbf{x}^*)\| + \left\|\frac{\mathbf{u}^+(t)}{\int_0^t \kappa(\tau)\mathrm{d}\tau}\right\|\right).
\end{aligned}
$$
$$(16)$$

When there exists a constant $\gamma < 1$ such that $\|\mathbf{W}\|_2 \leq \gamma V_{th}$, we have:

$$
\begin{aligned}
& \|\hat{\mathbf{a}}(t) - \hat{\mathbf{a}}(t-\Delta t_d)\| \\
\leq \quad & \left\|\text{ReLU}\left(\frac{1}{V_{th}}(\mathbf{W}\hat{\mathbf{a}}(t-\Delta t_d) + \mathbf{F}\mathbf{x}^* + \mathbf{b})\right) - \text{ReLU}\left(\frac{1}{V_{th}}(\mathbf{W}\hat{\mathbf{a}}(t-2\Delta t_d) + \mathbf{F}\mathbf{x}^* + \mathbf{b})\right)\right\| \\
& + \frac{1}{V_{th}}\left(\left\|\frac{\int_{t-\Delta t_d}^t \kappa(\tau)\mathrm{d}\tau}{\int_0^t \kappa(\tau)\mathrm{d}\tau}\mathbf{W}\hat{\mathbf{a}}(t-\Delta t_d)\right\| + \|\mathbf{F}(\hat{\mathbf{x}}(t) - \mathbf{x}^*)\| + \left\|\frac{\mathbf{u}^+(t)}{\int_0^t \kappa(\tau)\mathrm{d}\tau}\right\|\right. \\
& \left. + \left\|\frac{\int_{t-2\Delta t_d}^{t-\Delta t_d} \kappa(\tau)\mathrm{d}\tau}{\int_0^{t-\Delta t_d} \kappa(\tau)\mathrm{d}\tau}\mathbf{W}\hat{\mathbf{a}}(t-2\Delta t_d)\right\| + \|\mathbf{F}(\hat{\mathbf{x}}(t-\Delta t_d) - \mathbf{x}^*)\| + \left\|\frac{\mathbf{u}^+(t-\Delta t_d)}{\int_0^{t-\Delta t_d} \kappa(\tau)\mathrm{d}\tau}\right\|\right) \\
\leq \quad & \gamma\|\hat{\mathbf{a}}(t-\Delta t_d) - \hat{\mathbf{a}}(t-2\Delta t_d)\| \\
& + \frac{1}{V_{th}}\left(\left\|\frac{\int_{t-\Delta t_d}^t \kappa(\tau)\mathrm{d}\tau}{\int_0^t \kappa(\tau)\mathrm{d}\tau}\mathbf{W}\hat{\mathbf{a}}(t-\Delta t_d)\right\| + \|\mathbf{F}(\hat{\mathbf{x}}(t) - \mathbf{x}^*)\| + \left\|\frac{\mathbf{u}^+(t)}{\int_0^t \kappa(\tau)\mathrm{d}\tau}\right\|\right. \\
& \left. + \left\|\frac{\int_{t-2\Delta t_d}^{t-\Delta t_d} \kappa(\tau)\mathrm{d}\tau}{\int_0^{t-\Delta t_d} \kappa(\tau)\mathrm{d}\tau}\mathbf{W}\hat{\mathbf{a}}(t-2\Delta t_d)\right\| + \|\mathbf{F}(\hat{\mathbf{x}}(t-\Delta t_d) - \mathbf{x}^*)\| + \left\|\frac{\mathbf{u}^+(t-\Delta t_d)}{\int_0^{t-\Delta t_d} \kappa(\tau)\mathrm{d}\tau}\right\|\right),
\end{aligned}
$$
$$(17)$$

and

$$\frac{1}{V_{th}}\|\mathbf{W}(\hat{\mathbf{a}}(t) - \hat{\mathbf{a}}(t-\Delta t_d))\| \leq \gamma\|\hat{\mathbf{a}}(t) - \hat{\mathbf{a}}(t-\Delta t_d)\|. \quad (18)$$

Since $\mathbf{u}^+(t)$ is bounded by a constant $c$, $\frac{\int_{t-\Delta t_d}^t \kappa(\tau)\mathrm{d}\tau}{\int_0^t \kappa(\tau)\mathrm{d}\tau} \to 0$ and $\hat{\mathbf{x}}(t) \to \mathbf{x}^*$, there exists a constant $c'$ and $M$ such that when $t > M$, the following holds:

$$\|\hat{\mathbf{a}}(t) - \hat{\mathbf{a}}(t-\Delta t_d)\| \leq \gamma\|\hat{\mathbf{a}}(t-\Delta t_d) - \hat{\mathbf{a}}(t-2\Delta t_d)\| + c'. \quad (19)$$

Thus $\|\hat{\mathbf{a}}(t) - \hat{\mathbf{a}}(t-\Delta t_d)\|$ is bounded by $\frac{c'}{1-\gamma}$ when $t$ is large enough. Plugging this and Eq. (18) into Eq. (16), we get that the random error is bounded by a constant related with $c, V_{th}, \tau_m, \gamma$. This leads to Proposition 1.

**Proposition 1.** *If the weighted average inputs converge to an equilibrium point $\hat{\mathbf{x}}(t) \rightarrow \mathbf{x}^*$, and there exists constant $c$ and $\gamma < 1$ such that $|\mathbf{u}_i^+(t)| \leq c, \forall i, t$ and $\|\mathbf{W}\|_2 \leq \gamma V_{th}$, then the weighted average firing rates $\hat{\mathbf{a}}(t)$ of FSNN with continuous LIF model gradually approximate an equilibrium point $\mathbf{a}^*$ with bounded random errors, which satisfies $\mathbf{a}^* = ReLU\left(\frac{1}{V_{th}}\left(\mathbf{W}\mathbf{a}^* + \mathbf{F}\mathbf{x}^* + \mathbf{b}\right)\right)$.*

Although there would be random error, the remaining membrane potential of the LIF model will gradually decrease if there is no positive input, which means the random error tend to be eliminated. We can still view the FSNN with LIF model as approximately solving the fixed-point equilibrium equation.

## E.2 Discrete Perspective

The discrete update equation of membrane potentials under LIF model is:
$$\mathbf{u}[t+1] = \lambda \mathbf{u}[t] + \mathbf{W}\mathbf{s}[t] + \mathbf{F}\mathbf{x}[t] + b - V_{th}\mathbf{s}[t+1]. \tag{20}$$

Define the weighted average firing rate as $\hat{\mathbf{a}}[t] = \frac{\sum_{\tau=1}^t \lambda^{t-\tau}\mathbf{s}[\tau]}{\sum_{\tau=1}^t \lambda^{t-\tau}}$ and the weighted average inputs as $\hat{\mathbf{x}}[t] = \frac{\sum_{\tau=0}^t \lambda^{t-\tau}\mathbf{x}[\tau]}{\sum_{\tau=0}^t \lambda^{t-\tau}}$, then through summation and consideration of the division of $\mathbf{u}[t+1]$, we have:

$$\hat{\mathbf{a}}[t+1] = \frac{1}{V_{th}}\left(\frac{\sum_{i=0}^{t-1}\lambda^i}{\sum_{i=0}^t \lambda^i}\mathbf{W}\hat{\mathbf{a}}[t] + \mathbf{F}\hat{\mathbf{x}}[t] + \mathbf{b} - \frac{\mathbf{u}[t+1]}{\sum_{i=0}^t \lambda^i}\right), \tag{21}$$

$$\hat{\mathbf{a}}[t+1] = \sigma\left(\frac{1}{V_{th}}\left(\frac{\sum_{i=0}^{t-1}\lambda^i}{\sum_{i=0}^t \lambda^i}\mathbf{W}\hat{\mathbf{a}}[t] + \mathbf{F}\hat{\mathbf{x}}[t] + \mathbf{b}\right)\right) - \frac{1}{V_{th}}\frac{\mathbf{u}^+[t+1]}{\sum_{i=0}^t \lambda^i}, \tag{22}$$

where $\sigma(x) = \begin{cases} 1, & x > 1 \\ x, & 0 \leq x \leq 1 \\ 0, & x < 0 \end{cases}$.

Proposition 2 is similarly derived as Proposition 1 by substituting the ReLU function with $\sigma$. We omit repetitive details here.

**Proposition 2.** *If the weighted average inputs converge to an equilibrium point $\hat{\mathbf{x}}[t] \rightarrow \mathbf{x}^*$, and there exists constant $c$ and $\gamma < 1$ such that $|\mathbf{u}_i^+[t]| \leq c, \forall i, t$ and $\|\mathbf{W}\|_2 \leq \gamma V_{th}$, then the weighted average firing rates $\hat{\mathbf{a}}[t]$ of FSNN with discrete LIF model gradually approximate an equilibrium point $\mathbf{a}^*$ with bounded random errors, which satisfies $\mathbf{a}^* = \sigma\left(\frac{1}{V_{th}}\left(\mathbf{W}\mathbf{a}^* + \mathbf{F}\mathbf{x}^* + \mathbf{b}\right)\right)$.*

## E.3 Multi-layer Structure

When the multi-layer structure is considered, the discrete update equation of membrane potentials are expressed as:
$$\begin{cases} \mathbf{u}^1[t+1] = \lambda\mathbf{u}^1[t] + \mathbf{W}^1\mathbf{s}^N[t] + \mathbf{F}^1\mathbf{x}[t] + \mathbf{b}^1 - V_{th}\mathbf{s}^1[t+1], \\ \mathbf{u}^{l+1}[t+1] = \lambda\mathbf{u}^{l+1}[t] + \mathbf{F}^{l+1}\mathbf{s}^l[t+1] + \mathbf{b}^{l+1} - V_{th}\mathbf{s}^{l+1}[t+1], \quad l = 1, 2, \cdots, N-1 \end{cases} \tag{23}$$

Define the weighted average firing rates of different layers as $\hat{\mathbf{a}}^l[t] = \frac{\sum_{\tau=1}^t \lambda^{t-\tau}\mathbf{s}^l[\tau]}{\sum_{\tau=1}^t \lambda^{t-\tau}}$ and the weighted average inputs as $\hat{\mathbf{x}}[t] = \frac{\sum_{\tau=0}^t \lambda^{t-\tau}\mathbf{x}[\tau]}{\sum_{\tau=0}^t \lambda^{t-\tau}}$, then through summation and consideration of the division of $\mathbf{u}^l[t+1]$, we have:

$$\begin{cases} \hat{\mathbf{a}}^1[t+1] = \sigma\left(\frac{1}{V_{th}}\left(\frac{\sum_{i=0}^{t-1}\lambda^i}{\sum_{i=0}^t \lambda^i}\mathbf{W}^1\hat{\mathbf{a}}^N[t] + \mathbf{F}^1\hat{\mathbf{x}}[t] + \mathbf{b}^1\right)\right) - \frac{1}{V_{th}}\frac{\mathbf{u}^{1+}[t+1]}{\sum_{i=0}^t \lambda^i}, \\ \hat{\mathbf{a}}^{l+1}[t+1] = \sigma\left(\frac{1}{V_{th}}\left(\mathbf{F}^{l+1}\hat{\mathbf{a}}^l[t+1] + \mathbf{b}^{l+1}\right)\right) - \frac{1}{V_{th}}\frac{\mathbf{u}^{l+1+}[t+1]}{\sum_{i=0}^t \lambda^i}, \quad l = 1, \cdots, N-1. \end{cases} \tag{24}$$

With similar techniques in the proof of Theorem 3 and Proposition 1, we can derive the Proposition 3 as the following. We omit repetitive details here.

**Proposition 3.** *If the weighted average inputs converge to an equilibrium point $\hat{\mathbf{x}}[t] \to \mathbf{x}^*$, and there exists constant $c$ and $\gamma < 1$ such that $|\mathbf{u}_i^{l^+}[t]| \le c, \forall i, l, t$ and $\|\mathbf{W}^1\|_2 \|\mathbf{F}^N\|_2 \cdots \|\mathbf{F}^2\|_2 \le \gamma V_{th}^N$, then the weighted average firing rates $\hat{\mathbf{a}}^l[t]$ of multi-layer FSNN with discrete LIF model gradually approximate equilibrium points $\mathbf{a}^{l^*}$ with bounded random errors, which satisfy $\mathbf{a}^{1^*} = f_1\left(f_N \circ \cdots \circ f_2(\mathbf{a}^{1^*}), \mathbf{x}^*\right)$ and $\mathbf{a}^{l+1^*} = f_{l+1}(\mathbf{a}^{l^*})$, where $f_1(\mathbf{a}, \mathbf{x}) = \sigma\left(\frac{1}{V_{th}}(\mathbf{W}^1\mathbf{a} + \mathbf{F}^1\mathbf{x} + \mathbf{b}^1)\right), f_{l+1}(\mathbf{a}) = \sigma\left(\frac{1}{V_{th}}(\mathbf{F}^{l+1}\mathbf{a} + \mathbf{b}^{l+1})\right).$*

# F    Implementation Details

In this section, we describe the details for training our model. We will first introduce some operations in our model including restriction on the spectral norm and batch normalization, and then elaborate the training settings for the experiments.

## F.1    Restriction on Spectral Norm

As indicated in the theorems and propositions, a sufficient condition for the convergence of FSNN is $\|\mathbf{W}\|_2 \le \gamma V_{th}$ or $\|\mathbf{W}^1\|_2 \|\mathbf{F}^N\|_2 \cdots \|\mathbf{F}^2\|_2 \le \gamma V_{th}^N$, where $\gamma < 1$. To ensure the convergence of the forward SNN computation and stabilize training, we propose to restrict the spectral norm of the feedback connection weight matrix. Specifically, we re-parameterize $\mathbf{W}$ as:

$$\mathbf{W} = \alpha \frac{\mathbf{W}}{\|\mathbf{W}\|_2}, \tag{25}$$

where $\alpha$ is a learnable parameter and will be clipped in the range of $[-c, c]$ ($c$ is a constant), and the spectral norm $\|\mathbf{W}\|_2$ is similarly computed as the implementation of Spectral Normalization [6]. In experiments, we will set $V_{th} = 2$ and $c = 1$, and for the multi-layer structure, we only restrict the spectral norm of feedback connection weight $\mathbf{W}^1$. It works well in practice and the convergence is illustrated in Section 5.4 and Section G.2.

## F.2    Batch Normalization

Batch normalization (BN) [3] is a commonly adopted technique in ANNs, which accelerates the training by reducing the internal covariate shift and improves performance as well. For a $d$-dimensional data $x = \left(x^{(1)} \cdots x^{(d)}\right)$, BN normalizes and transforms the data as:

$$\hat{x}^{(k)} = \gamma^{(k)} \frac{x^{(k)} - \mathrm{E}[x^{(k)}]}{\sqrt{\mathrm{Var}[x^{(k)}]}} + \beta^{(k)}, \tag{26}$$

where $\mathrm{E}[x^{(k)}]$ and $\mathrm{Var}[x^{(k)}]$ are statistics over the training data set, and $\gamma^{(k)}, \beta^{(k)}$ are learnable parameters.

Note that when the statistics are fixed, BN is a simple linear transformation, and BN after a linear layer can be absorbed into the parameters of this layer. For example, for the linear operation $y = Wx + b$ (suppose $y$ is one-dimensional for simplicity), let $e, v$ denote the expectation and variance of $y$, then $\hat{y} = \mathrm{BN}(y)$ is equivalent as a new linear operation $\hat{y} = \widetilde{W}x + \widetilde{b}$, where $\widetilde{W} = \frac{\gamma}{\sqrt{v}}W, \widetilde{b} = b - \frac{\gamma e}{v} + \beta$. Therefore, adding BN with fixed statistics after a convolution or fully-connected layer will not influence the properties of SNNs and the conclusions for equilibrium convergence.

We add BN after each linear operation except the feedback layer, in the context of the fixed-point equilibrium equation. For example, for the single-layer FSNN whose equation is $\mathbf{a}^* = \mathrm{ReLU}\left(\frac{1}{V_{th}}(\mathbf{W}\mathbf{a}^* + \mathbf{F}\mathbf{x}^* + \mathbf{b})\right)$, we add BN after $\mathbf{F}\mathbf{x}^*$; and for the multi-layer FSNN, we add BN after $\mathbf{F}^1\mathbf{x}^*$ and $\mathbf{F}^{l+1}\mathbf{a}^{l^*}$.

During forward SNN computation, the statistics of BN operations are fixed, i.e. we set BN into the 'eval' mode which uses the previously calculated statistics; and during backward gradient calculation, since it is decoupled from the forward computation (that means we will construct an additional computational graph for it), we can follow the common setting of BN to leverage the mini-batch estimated statistics and the overall statistics are updated, i.e. we set BN into the 'train' mode in

this computational graph. The statistics are for the (weighted) average inputs or firing rates. Since the estimation of statistics for the forward SNN computation may be inaccurate in the first several iterations, we will use a warmup for the learning rate to alleviate this problem.

## F.3 Training Settings

### F.3.1 Datasets

We conduct experiments on MNIST [5], Fashion-MNIST [10], N-MNIST [7], CIFAR-10 and CIFAR-100 [4].

**MNIST** MNIST is a dataset of handwritten digits with 10 classes, which is composed of 60,000 training samples and 10,000 testing samples. Each sample is a $28 \times 28$ grayscale image. We normalize the inputs based on the global mean and standard deviation, and convert the pixel value into a real-valued input current at every time step. No data augmentation is applied.

The licence of MNIST is the MIT License. The MNIST database is constructed from NIST's Special Database 3 and Special Database 1 which contain binary images of handwritten digits [5]. The data does not contain personally identifiable information or offensive content since it only consists of handwritten digits.

**Fashion-MNIST** Fashion-MNIST is a dataset similar to MNIST and contains $28 \times 28$ grayscale images of clothing items. We use the same preprocessing as MNIST.

The licence of Fashion-MNIST is the MIT License. The data of Fahion-MNIST is collected from the photographs of fashion products on the assortment on Zalando's website [5]. The data does not contain personally identifiable information or offensive content since it only consists of 10 kinds of fashion products.

**N-MNIST** N-MNIST is a neuromorphic dataset that is converted from MNIST by a Dynamic Version Sensor (DVS). It consists of spike trains triggered by the intensity change of pixels when DVS scans the static MNIST images along given directions. Since the intensity can either increase or decrease, there are two channels corresponding to ON- and OFF-event spikes. And the pixel dimension is expanded to $34 \times 34$ due to the relative shift of images. Therefore, each sample is a spike train pattern with the size of $34 \times 34 \times 2 \times T$, where $T$ is the temporal length. The original data record $300ms$ with the resolution of $1\mu s$. We follow the prepossessing of [11] to reduce the time resolution by accumulating the spike train within every $3ms$, and we will use the first 30 time steps.

The license of N-MNIST is the Creative Commons Attribution-ShareAlike 4.0 license. The data is converted from MNIST and does not contain personally identifiable information or offensive content.

**CIFAR-10** CIFAR-10 is a dataset of color images with 10 classes of objects, which is composed of 50,000 training samples and 10,000 testing samples. Each sample is a $32 \times 32 \times 3$ color image. We normalize the inputs based on the global mean and standard deviation, and apply random cropping and horizontal flipping for data augmentation. The input pixel value is converted to a real-valued input current at every time step as well.

**CIFAR-100** CIFAR-100 is a dataset similar to CIFAR-10 except that there are 100 classes of objects. It also consists of 50,000 training samples and 10,000 testing samples. We use the same preprocessing as CIFAR-10.

The license of CIFAR-10 and CIFAR-100 is the MIT License. The data are labeled subsets of the 80 million tiny images datasets (collected from the web), which are labeled by students [4]. The data does not contain personally identifiable information or offensive content, which is checked by the classes and image samples.

### F.3.2 Training Hyperparameters

For all our SNN models, we set $V_{th} = 2$. For the LIF model, we set $\lambda = 0.95$ for MNIST, Fashion-MNIST and N-MNIST, while $\lambda = 0.99$ for CIFAR-10 and CIFAR-100.

We train all our models by SGD with momentum for 100 epochs. We set the momentum as 0.9, the batch size as 128, and the initial learning rate as 0.05. For MNIST, Fashion-MNIST, and N-MNIST, the learning rate is decayed by 0.1 every 30 epochs, while for CIFAR-10 and CIFAR-100, it is decayed by 0.1 at the 50th and 75th epoch. We also apply linear warmup for the learning rate in the first 400 iterations for CIFAR-10 and CIFAR-100. We set the weight decay as $5 \times 10^{-4}$, and apply the variational dropout as in [1, 2] with dropout rate as 0.2. For MNIST, Fashion-MNIST, CIFAR-10, and CIFAR-100, we solve the implicit differentiation by the Broyden's method proposed in [2] with the threshold as 30. For N-MNIST, we solve the implicit differentiation by the fixed-point update scheme indicated in Section 3.2 for 30 iterations, and the update scheme is modified as $x^T = \frac{1}{2} \left( x^T + (J_{f_\theta}^T|_{a^*})x^T + \left( \frac{\partial \mathcal{L}(a^*)}{\partial a^*} \right)^T \right)$ for acceleration. The initialization of parameters follows [9], which first samples the weight parameters from the standard uniform distribution and then normalize them for each output dimension. All experiments are repeated five times and we report the mean, standard deviation, and the best results.

The code implementation is based on the PyTorch framework [8], and experiments are carried out on one NVIDIA GeForce GTX 1080 GPU or one NVIDIA GeForce RTX 3090 GPU.

# G   Additional Experiment Results

## G.1   Comparison between IF and LIF Model on CIFAR-10 and CIFAR-100

In this subsection, we supplement the comparison results of IF and LIF model on CIFAR-10 and CIFAR-100, as shown in Table 1. It shows that the LIF model has similar performance compared with the IF model, and slightly outperforms the IF model in most cases, especially when the number of time steps is small. This also accords with the results on MNIST, Fashion-MNIST, and N-MNIST. The possible reason is that the LIF model leverages temporal information of spike trains by encoding weighted average firing rates. While each spike contributes equally to the average firing rate of the IF model and thus the precision of firing rates is only $\frac{1}{T}$, the weight for a spike of the LIF model is different at time steps (the weight is $\lambda^{T-t}$), and therefore the weighted average firing rates could encode more information with the same amount of time steps. When there is a relatively small number of time steps, the convergence errors of IF and LIF model would be similar and will not significantly affect the results. So the LIF model with temporal information may perform slightly better.

Table 1: Comparison results of IF and LIF Model on CIFAR-10 and CIFAR-100.

| CIFAR-10 | | | | |
|---|---|---|---|---|
| Network structure | Time steps | Model | Mean±Std | Best |
| AlexNet-F | 30 | IF | 91.73%±0.13% | 91.85% |
| | | LIF | 91.74%±0.09% | 91.92% |
| AlexNet-F | 100 | IF | 92.25%±0.27% | 92.53% |
| | | LIF | 92.03%±0.07% | 92.15% |
| CIFARNet-F | 30 | IF | 91.94%±0.14% | 92.12% |
| | | LIF | 92.08%±0.15% | 92.23% |
| CIFARNet-F | 100 | IF | 92.33%±0.15% | 92.57% |
| | | LIF | 92.52%±0.17% | 92.82% |

[1] AlexNet-F: 96C3s-256C3-384C3s-384C3-256C3 (F96C3u)
[2] CIFARNet-F: 128C3s-256C3-512C3s-1024C3-512C3 (F128C3u)

| CIFAR-100 | | | | |
|---|---|---|---|---|
| Network structure | Time steps | Model | Mean±Std | Best |
| CIFARNet-F | 30 | IF | 71.56%±0.31% | 72.10% |
| | | LIF | 71.72%±0.22% | 72.03% |
| CIFARNet-F | 100 | IF | 73.07%±0.21% | 73.43% |
| | | LIF | 72.98%±0.13% | 73.12% |

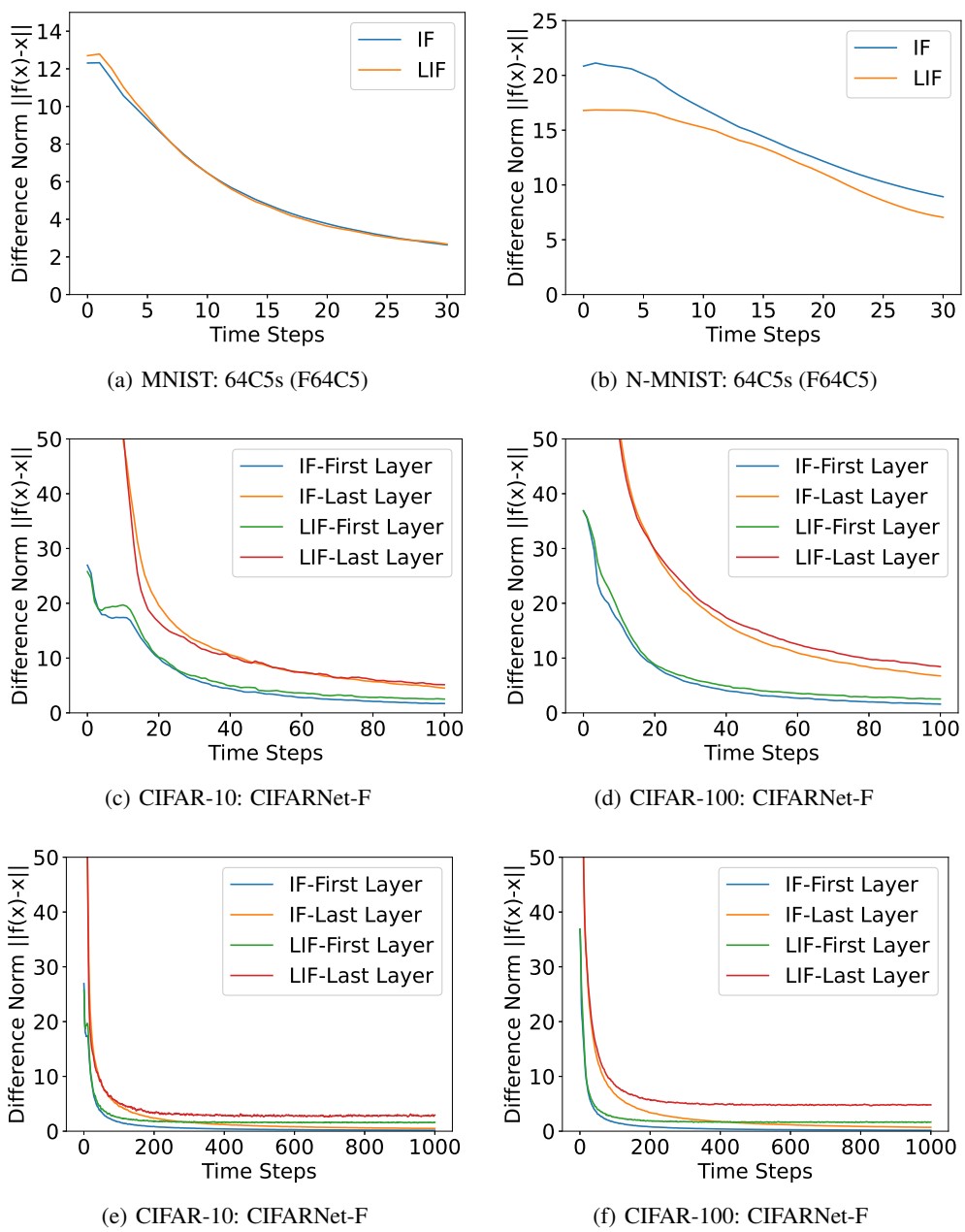

(a) MNIST: 64C5s (F64C5)

(b) N-MNIST: 64C5s (F64C5)

(c) CIFAR-10: CIFARNet-F

(d) CIFAR-100: CIFARNet-F

(e) CIFAR-10: CIFARNet-F

(f) CIFAR-100: CIFARNet-F

Figure 2: Convergence to the equilibrium of different models.

### G.2 Convergence to Equilibrium

In this subsection, we supplement the results of convergence to the equilibrium state for more datasets and different scales of time steps. Figure 2 illustrates the convergence information on MNIST and N-MNIST with the same network structure under 30 time steps, as well as the convergence information on CIFAR-10 and CIFAR-100 with the same network structure under 100 and 1000 time steps. Since the precision of firing rates under relatively few time steps is limited, there would be errors caused by the precision. And the more the time steps are, the less the error should be. The difference norm decreases with time steps under all settings, demonstrating the convergence to the equilibrium state with the fixed-point equation. For N-MNIST, since the inputs are neuromorphic spikes rather than static images at each time step (and there lacks valid information in the first few time steps), the convergence is slower than on MNIST. Despite this, the (weighted) average firing rates do gradually approach the equilibrium as the difference norm decreases, and the training based on the implicit differentiation can work well as shown in the accuracy results. For the multi-layer structure, the convergence error of the last layer would be larger than the first layer. For the LIF model, since there would be random errors as indicated in the propositions, the convergence error would be larger than the IF model at most time. Nevertheless, when the number of time steps is small, the difference is not apparent. When the number of time steps comes to 1000, it shows that the error of IF model would continuously decrease, while the error of LIF model may stay in a bounded range, which should be caused by the random error. Despite these convergence errors, the accuracy results demonstrate that the exact precision is not necessary for effective training based on the implicit differentiation, and we can actually achieve satisfactory results with a small number of time steps.

## H  Influence of Time Steps

In this subsection, we further study the influence of time steps, i.e. how good the convergence to equilibrium states needs to be for effective prediction and training. We first study the performance of a pretrained model under different time steps. The results are in Table 2. It shows that the accuracy will gradually decrease as the time step decreases, and when the time step is 5, the classification totally fails. We also briefly analyze the total average firing rate under these conditions as in Table 2, and it shows that the firing rate is very low when the time step is small. So the accuracy drop may also be partly due to the insufficient spikes.

Table 2: Performance of a pretrained model under different time steps. The model is trained on CIFAR-10 with AlexNet-F structure and LIF model, and the original time step is 30.

| Time steps | Accuracy | Total average firing rate |
|---|---|---|
| 5 | 10.67% | 0.0016 |
| 10 | 74.39% | 0.0046 |
| 15 | 87.07% | 0.0059 |
| 20 | 90.16% | 0.0063 |
| 25 | 91.33% | 0.0065 |
| 30 | 91.82% | 0.0066 |

Then to study the influence of time steps on training, we train and test our model with only 5 time steps. The results are in Table 3. It shows that the training does not fail when the number of time steps is 5, but there would be a significant performance drop, and the accuracy would decrease and fluctuate in the latter part of training. It is probably because the gradient calculated by implicit differentiation could still be a descent direction though not exact, and in the latter part, it may not be a descent direction so the accuracy cannot be further improved.

Table 3: Performance of training the model under different time steps. The model is trained on CIFAR-10 with AlexNet-F structure and LIF model.

| Time steps | Accuracy |
|---|---|
| 5 | 83.09% |
| 30 | 91.74%±0.09% |

# I Discussion of Limitations and Social Impacts

This work mainly focuses on training feedback spiking neural networks for inputs that are convergent in the context of average accumulated signals, as indicated in the assumptions in the theorems. This holds for common pattern recognition tasks and common visual tasks, e.g. image classification, whose inputs are static images or the alternative neuromorphic version with spikes. While for other types of varying inputs, e.g. speech, it may require additional efforts to consider the definition and utilization of equilibrium with time. One practically plausible method is to flatten the inputs to treat the original time dimension as the channel dimension, and feed such data to the model at each 'time step'. In this way, our theorems and method still hold. But the definition of 'time step' in this method is not the true time, which may lack the biological plausibility and increase the computational requirements. An interesting future work is to generalize the methodology to varying inputs.

As for social impacts, since this work focuses only on training methods for spiking neural networks, there is no direct negative social impact. And we believe that the development of successful energy-efficient SNN models could broader its applications and alleviate the huge energy consumption by ANNs. Besides, understanding and improving the training of biologically plausible SNNs may also contribute to the understanding of our brains and bridge the gap between biological neurons and successful deep learning.