# OpenReview forum: "Training Feedback Spiking Neural Networks by Implicit Differentiation on the Equilibrium State"
_NeurIPS.cc/2021/Conference — NeurIPS 2021 Spotlight_

### Official Review · Reviewer_Gsxo · 2021-07-07

**Rating:** 6
**Confidence:** 4

**Summary:**

This paper presented a training method for feedback spiking neural networks to tackle existing training problems with backpropagation framework. Experiments were conducted on five image datasets, MNIST, Fashion-MNIST, N-MNIST, CIFAR-10 and CIFAR-100, superior performances were claimed.

**Limitations And Societal Impact:**

The writing is clear and the motivation is clarified clearly. Besides, the theoretical grounding and experimental evaluation are not sufficient to show their originality and significance. Here are some of the suggestions:
1)	I would like to see ablation studies for the proposed training method, the traditional backpropagation framework refers as the baseline.
2)	How to deal with the different types of inputs (e.g., bio-medical signals or speech)? It would be valuable to discuss it and present your solutions in this paper.
The citation seems a bit disordered.

**Main Review:**

The idea of this paper is novel, training a robust deep SNN is very critical in brain-inspired learning field. From the prespective of equilibrium of NNs is impressive, however, I didn't see more details about the equilibrium of SNN. Based on BP, the authors proposed IDE method to try to train a deep SNN. Experimental results evaluated on MNIST and CIFAR showed that the proposed model could achieve good classification performance on those datasets with fewer time steps.

However, there are some missing descriptions. What is the difference when you applied IF or LIF, and which situation did you prefer IF of LIF? In section 3.2, when you choose the fixed-point operation, how can you guarantee the convergence？

The experiments conducted on MNIST or CIFAR based were not enough when you proposed a BP-based model to train a deep SNN. I prefer to see the results from ImageNet datasets or analysis of energy consumption.



**Time Spent Reviewing:**

2

---

> ### Author Response · Authors · 2021-08-10
> **Response to Reviewer Gsxo**
>
> Thank you for your comments and for viewing our work novel and impressive. We try our best to address your concerns as follows.
>
> 1. “I didn’t see more details about the equilibrium of SNN”. We respectively disagree that there are no more details. The most part of Section 4 is to derive the equilibrium states for feedback spiking neural networks for both IF and LIF models under both continuous and discrete views as well as with multi-layer structure, and all theorems and propositions are to derive the equilibrium. So would you please clarify what other kind of details you mean here?
>
> 2. “About difference between IF and LIF, and the preference”. IF and LIF are both commonly used spiking neuron models with similar effects, and the difference when applying them includes their forward computation and definition of firing rates. In Section 3.1, we have described the difference in the forward computation: there will be an additional leaky term \lambda for LIF, as in line 135-138. In Section 4.1, we have described that the IF model will consider the average firing rate while the LIF model will consider the weighted average firing rate related to the leaky term, as in line 164-165, 184-188, 196-197, 208-211. And for the gradient computation based on implicit differentiation, since the equilibrium equation for the IF (w.r.t average firing rate) and for the LIF (w.r.t weighted average firing rate) has a same form, so the calculation for implicit differentiation is the same, and we describe it together in Section 4.2. Both IF and LIF are commonly used models, so we do not have a prior preference, and we have studied the comparison between them in the experiments, as shown in Sections 5 and E.1 in the appendix. The results show that LIF slightly outperforms IF most of the time.
>
> 3. “How to guarantee the convergence when choosing the fixed-point operation”. In line 149, we have described that the fixed-point update scheme will converge with linear convergence rate as long as $\lVert J_{f_{\theta}}^T\vert_{a^*} \rVert< 1$, which is an existing conclusion in optimization. And since we will restrict the spectral norm of the weight matrix to satisfy the assumptions in the theorems and propositions, which is described in Section D.1 in the appendix, it will also satisfy the above condition for the convergence of the fixed-point update.
>
> 4. “Experiments on MNIST or CIFAR are not enough, prefer ImageNet or analysis of energy consumption”. We respectfully disagree that MNIST and CIFAR are not enough and ImageNet is essential. The common practice to evaluate direct training methods for SNNs is on MNIST or CIFAR [14, 16, 21, 36, 39, 40, 44]. To our best knowledge, no result on ImageNet is reported for directly trained SNNs except only one recent paper from AAAI 2021 [45], because of the difficulty in direct SNN training as described in the introduction. It would be an interesting future work to extend to ImageNet, but currently MNIST and CIFAR are enough for verifying the novel training method for SNNs.
> As for energy consumption, we supplement some results about average firing rates as below (CIFAR-10 dataset, AlexNet-F structure, 30 time steps).
> | Layer | Average firing rate (IF model) | Average firing rate (LIF model) |
> | :----: | :----: | :----: |
> | layer1 | 0.0345 | 0.0166 |
> | layer2 | 0.0041 | 0.0039 |
> | layer3 | 0.0025 | 0.0024 |
> | layer4 | 0.0008 | 0.0008 |
> | layer5 | 0.0399 | 0.0177 |
> | total  | 0.0119 | 0.0066 |
>
>     It shows that the firing rate of our trained model is only about or less than 1%, which is smaller than previous work such as TSSL-BP [44] whose firing rate is roughly 9.86%, so our trained model has fewer spikes. Since the energy consumption of SNNs is proportional to the number of spikes, it indicates less energy consumption of our model. As for the specific numerical value, it is related to hardware support and could be future work.
>
> 5. “Ablation studies to compare with traditional backpropagation framework”. As described in the introduction, the traditional BP framework for ANN cannot be directly generalized to SNN training due to non-differentiability, and some previous efforts have been made to imitate the BPTT framework with surrogate gradients or computing gradients w.r.t the spiking time. If your ‘traditional’ means these BPTT-like methods, first, many of these methods such as STBP [39, 40] do not release their code, so we simply report the results in their papers; second, some of these methods such as [14, 44] rely on the complex synaptic and current model, which cannot be directly applied to the simple current model in our work and other work like [39, 40], so we just report their results as well.
>
>     Still, to respond to your suggestions, we report the results by applying STBP [39, 40] re-implemented by ourselves with the same setting as ours. The experiment is on CIFAR-10 with AlexNet-F structure and LIF model, all hyper-parameters are the same.
> Our response to Reviewer u6bR reports the memory cost comparison under this setting, and we also report it here. The accuracy and GPU memory costs are below:
> | Method | Time steps | Accuracy | GPU memory |
> | ---- | :----: | :----: | :----: |
> | IDE (ours) | 30 | 91.74±0.09% | 2.8G |
> | STBP | 30 | 87.18% | 11G |
> | IDE (ours) | 100 | 92.03±0.07% | 2.8G |
> | STBP | 100 | / | GPU out of memory |
>
>     It shows that our method can significantly reduce the memory cost as we avoid backpropagating along the computational graph unfolded along time, and we can train with 100 time steps using the same memory cost while STBP requires exceeding memory and fails to train under this setting. Also, the accuracy of our method is significantly better than STBP.
>
>     Besides, we also study the average firing rate of the same model trained by our method and STBP. The results are below (30 time steps):
> | Layer | Average firing rate (STBP) | Average firing rate (IDE, ours) |
> | :----: | :----: | :----: |
> | layer1 | 0.0190 | 0.0166 |
> | layer2 | 0.0082 | 0.0039 |
> | layer3 | 0.0113 | 0.0024 |
> | layer4 | 0.0055 | 0.0008 |
> | layer5 | 0.0108 | 0.0177 |
> | total  | 0.0102 | 0.0066 |
>
>     It shows that there are different firing patterns and the model trained by our method has sparser spikes, which indicates that lower energy consumption is required. It demonstrates the superior result of our method and great potential for energy-efficient neuromorphic implementation as well.
>
> 6. “About different types of inputs”. Our method is suitable for static or convergent inputs or the alternative neuromorphic version with spikes. As for non-stationary input, we have discussed it in Section F in the appendix. There could be some practically plausible but costly solutions, i.e. flattening the inputs to treat the original time dimension as the channel dimension, just as deep equilibrium models do [3], and it would be an interesting future work to efficiently generalize our methodology to varying inputs.
>
> Finally, we would again emphasize the contribution of our work. We are the first to theoretically derive the equilibrium states with a fixed-point equation for FSNNs and propose a novel direct training method for SNNs based on implicit differentiation, which avoids common SNN training problems including non-differentiability and large memory costs. Moreover, the method is connected with the biologically plausible Hebbian learning rule and would pave a solid way for improving on-chip neuromorphic learning. Extensive experiments demonstrate superior results with fewer neurons and parameters in a small number of time steps, and with very sparse spikes, which show significant potential for energy-efficient neuromorphic computation. We think it is sufficient to show the originality and significance of our work. We have tried our best to address your concerns, and if you have new questions we could discuss then.

---

> > ### Comment · Reviewer_Gsxo · 2021-08-19
> > **My new response to authors**
> >
> > I am satisfied to see that the authors analyzed the average firing rate between IF and LIF as I suggested, so I raise my score from 4 to 6, I am happy to see this paper would be accepted.

---

### Official Review · Reviewer_u6bR · 2021-07-09

**Rating:** 7
**Confidence:** 3

**Summary:**

The paper proposes a new training method for SNNs that relies on decoupling the forward propagation from the backward by viewing the SNNs as solving a fixed-point equilibrium equation and exploiting implicit differentiation results to compute the gradients. The theoretical contribution is to derive the equilibrium states for the fixed-point equations that the SNNs are solving, based on which the implicit differentiation technique can be leveraged. The practical contribution is a training algorithm that avoids the problems of non-differentiability in SNNs and achieves state-of-the-art performance on many datasets.


**Limitations And Societal Impact:**

The proposed training algorithm for SNN relies on equilibrium states on firing rates for each layer of the network. This limits the SNN for non-stationary inputs (time-series data that changes over time, for example, audio and video) where such equilibrium states may not exist in the network.

**Main Review:**

The paper proposes a new training method for SNNs that relies on decoupling the forward propagation from the backward by viewing the SNNs as solving a fixed-point equilibrium equation and exploiting implicit differentiation results to compute the gradients. The theoretical contribution is to derive the equilibrium states for the fixed-point equations that the SNNs are solving, based on which the implicit differentiation technique can be leveraged. The practical contribution is a training algorithm that avoids the problems of non-differentiability in SNNs and achieves state-of-the-art performance on many datasets.

Highlights:

1. The most important contribution definitely is a training algorithm that does not rely on the exact forward procedure. The SNN literature is full of studies trying to overcome the issue of non-differentiability through pseudo-grad, or large memory requirements using simpler spiking models. There are very few studies that avoid these problems altogether and this is one of them.

2. The authors provide rigorous proof of the equilibrium states that the SNNs converge to under both continuous and discrete dynamics. As a result, one can leverage implicit differentiation models to compute the gradients w.r.t. the fixed point equations, thus avoiding the need for a backward procedure that relies on the forward procedure.

3. Based on 1, the authors propose a training algorithm for SNNs with the assumption that the average firing rates of the SNNs follow the fixed-point equations. The loss and gradients of the loss become straightforward to compute.
While not completely obvious at first, the authors provide a qualitative study of how the learning rule is similar to the Hebbian learning rule, thereby imparting it some degree of biological plausibility.

4. Experiments on many datasets, including both small benchmark datasets and large image datasets, show that the algorithm can obtain state-of-the-art performance with a much less number of network parameters and timesteps. This demonstrates the practical effectiveness of the method.



Concerns:

1. From equation 10, it doesn’t seem like the network is weight-tied and does not have input injection at each layer like in [1]. As a result, acc. to Theorem 3, the equilibrium point does not directly depend upon the inputs which might affect its stability. That’s not the case in [1] from which the method seems to have been inspired. Can the authors resolve this discrepancy, and conduct additional experiments on weight-tied deep networks?

2. How practical are the assumptions in Theorem 1? In particular, how does the fact that the membrane potentials and network weights need to be bounded affect the initialization of these hyperparameters? This is hinted at in the appendix, but I think it deserves a discussion in the main text. Moreover, why is alpha in equation 25 in appendix D a learnable parameter?

3. Is it fair to characterize the learning algorithm as rate-based? If that is the case, and combined with the fact that the inputs need to be convergent, how does the algorithm exploit the temporal capabilities of the SNNs? Is an event-based version of this algorithm possible?



Minor Comments:

1. In the introduction, the authors mention that “SNNs naturally compute with multiple time steps, which inherently supports feedback computation”. Yet, authors include a feedback connection in their architecture. Can the authors discuss what is the difference in the feedback information represented by these two mechanisms? In addition, it would be nice to have an ablation study of the network without feedback connections.

2. It seems like one advantage of the method is the decrease in training complexity over other ways of training the networks (such as STBP [2]). This might be due to the fact that backward computation relies on a simpler procedure than following the exact forward propagation. If that is the case, can the authors quantify the ease in training and include a short discussion.

3. It will be more clear if the authors can include algorithm pseudocode for the training algorithm.

[1] Shaojie Bai, J Zico Kolter, and Vladlen Koltun. Deep equilibrium models. In Advances in Neural Information Processing Systems, volume 32, pages 690–701, 2019.

[2] Yujie Wu, Lei Deng, Guoqi Li, Jun Zhu, and Luping Shi. Spatio-temporal backpropagation for training high-performance spiking neural networks. Frontiers in Neuroscience, 12:331, 2018.


**Time Spent Reviewing:**

30

---

> ### Author Response · Authors · 2021-08-10
> **Response to Reviewer u6bR**
>
> Thank you very much for appreciating our work and viewing our work novel. We address your concerns as follows.
>
> 1. About “weight-tied” and “input injection”. First, for deep equilibrium models, their weight-tied model corresponds to the iterative equation $x^{t+1} = f_{\theta}(x^t)$, where the weight-tied $f_{\theta}$ can be a block with several non-linear computation rather than only one layer, and their objective is to derive the equilibrium equation $x = f_{\theta}(x)$. The computation in Eq. (10) actually corresponds to the computation of $f_{\theta}$, and the calculation of this computation at each time step roughly corresponds to the “weight-tied” of $f_{\theta}$ (you may actually view those weight-tied models as a kind of recurrent neural network, which is easier to compare with our model, and our model is actually “weight-tied” at all time steps). Our multi-layer structure aims at enhancing the non-linearity of the equilibrium fixed-point equation (corresponding to the non-linearity of the block $f_{\theta}$), which is not related with whether weight-tied. Second, the equilibrium fixed-point equation in Theorem 3 does directly depend upon the inputs since it depends on the input $x$, and the calculation of $F^1x$ in the function $f_1(a, x)$ can be viewed as “input injection”. So there is no discrepancy here.
>
> 2. About the assumptions in Theorems. First, the bound of $u^+(t)$ is ensured by the definition and we have discussed it in line 168-175 and line 201-202 (the introduction of ReLU in Eq. (6) and $\sigma$ in Eq. (9) is to ensure the boundedness of $u^+(t)$). Second, to ensure the assumption that the spectral norm of the weight matrix is bounded, we restrict the spectral norm by re-parameterizing the weights in practice. Due to the space limit, we leave it to the appendix in Section D.1, and we will add some brief descriptions in the main text. As for the $\alpha$ in Section D.1, it is a re-parameterized parameter that is learnable while we control it in a range to restrict the spectral norm. Why it is learnable is because this enlarges the parameter space (otherwise the weight can only be on the sphere with the spectral norm being 1, ours enables the weight to be within this ball).
>
> 3. About whether the learning algorithm is rated-based and how to exploit the temporal capabilities. Our method is not exactly rate-based, but a combination of the rate and temporal information. For the LIF model, we define the weighted average firing rate based on the response kernel, which means that spikes at different time steps have different weights. This leverages the temporal capabilities of the SNNs. The event-based input is compatible as long as the (weighted) average input converges to an equilibrium point, as indicated in the theorems. For example, the experiment of N-MNIST just uses the event-based input. Our method is suitable for static/convergent inputs or the alternative neuromorphic version with spikes.
>
> 4. About feedback connection. The feedback computation we mentioned in the introduction is just feedback connection: SNNs naturally compute with multiple time steps so incorporating feedback connection does not have many costs compared with additionally unfolding along time as ANNs do. Feedback is essential for the dynamics and equilibrium, and if without feedback connections, it would degrade to direct functional mapping and the implicit differentiation would degrade to the explicit gradient. So we mainly focus on feedback spiking neural networks in our experiments.
>
> 5. About quantifying the ease in training. Since our method decouples the forward and backward computation, one significant advantage is that we can avoid large memory costs due to backpropagation along the computational graph unfolded along time such as STBP does. And our memory costs are agnostic to the simulation time steps while methods like STBP are proportional to the time steps. We present the GPU memory cost as below (AlexNet-F, CIFAR-10, LIF model, batch size 128):
> | Time steps | GPU memory (IDE, ours) | GPU memory (STBP) |
> | :----: | :----: | :----: |
> | 30 | 2.8G | 11G |
> | 100 | 2.8G | GPU out of memory (9G for batch size 32) |
>
> 6. About pseudocode. Thanks for pointing out this. We will add it in the revision as below.
>
> | Algorithm: One iteration of IDE training algorithm |
> | ---- |
> | Input: Initialized network parameters $\theta$; Input data $x$; Label $y$; Time steps T; Other hyperparameters; |
> | Output: Trained network parameters $\theta$.|
> |   Forward: |
> |  1: Simulate the SNN by T time steps with input $x$ based on Eq. (2) and calculate the final (weighted) average firing rate $a[T]$ |
> |  2: Calculate the output $o$ and the loss $L$ based on $o$ and $y$ |
> |  Backward: |
> |  1. Specify the fixed-point equation $a=f_{\theta}(a)$ of the equilibrium state (define $g_{\theta}(a)=f_{\theta}(a) - a$) |
> |  2: Calculate the gradients based on implicit differentiation |
> |   2.1: Solve the equation $\left(J_{g_{\theta}}^T\vert_{a[T]}\right)x^T+\left(\frac{\partial \mathcal{L}}{\partial a[T]}\right)^T=0$ by root-finding methods |
> |    2.2: Calculate the gradients $ \frac{\partial \mathcal{L}}{\partial \theta} = -\frac{\partial \mathcal{L}}{\partial a[T]} \left(J_{g_{\theta}}^{-1}\vert_{a[T]}\right) \frac{\partial f_{\theta}(a[T])}{\partial \theta}$ based on the solution and $\frac{\partial f_{\theta}(a[T])}{\partial \theta}$ |
> |  3: Update $\theta$ based on the gradient-based optimizer |
>
> 7. About non-stationary inputs. We have discussed this limitation in Section F in the appendix. There could be some practically plausible but costly solutions, i.e. flattening the inputs to treat the original time dimension as the channel dimension, just as deep equilibrium models do [3], and it would be an interesting future work to generalize such methodology to varying inputs.

---

### Official Review · Reviewer_W1bo · 2021-07-16

**Rating:** 8
**Confidence:** 4

**Summary:**

The authors propose a relatively novel way of training spiking networks by considering the equilibrium state of their firing rates in response to input, and training using implicit differentiation, as has recently been explored in deep equilibrium models and implicit layers. They consider leaky and perfect integrate-and-fire neurons, as well as single layers with recurrent feedback, and hierarchical layers with feedback from the last layer to the first. The authors prove that the dynamics do indeed converge to an equilibrium state, and demonstrate the performance of the model on several state-of-the-art datasets.

**Ethical Concerns:**

No concerns

**Limitations And Societal Impact:**

The authors did not include a broader impact statement.

**Main Review:**

The method of using implicit differentiation to train spiking networks appears novel and is very interesting! The description of the model dynamics and implicit differentiation is very clear (for the most part), and the results show that this method is indeed powerful and promising.

However, the main criticism of this work that I have is the lack of illustration of the model details. The paper jumps straight to the results, so while the authors show that the model works, no insights are gained beyond that. To aid the reader in understanding, it would be useful to have a couple of schematics illustrating the spiking network architecture for the single layer and hierarchical case. It would also be useful to show an example of the network responding to input over time, and comparing the spikes themselves with the estimates of the average firing rates in equations 6 and 9. This is a crucial part of getting implicit differentiation to work, so it seems important to show an example or two. Also, when thinking about biological plausibility and applications in neuromorphic hardware, the number of spikes fired is important, and it would be interesting if the authors can show some statistics on the average firing rate per neuron for the trained models.

Next, I have two more curiosities that I think are worth elaborating on in the paper a bit. One is about the stability of the equilibrium state during learning. The authors show that given particular conditions, the network will converge to equilibrium, but what’s stopping the network from violating these conditions during training? E.g., does the network ever evolve into limit cycle behavior? Second, while the authors show convergence in Fig 1, I am left wondering how good this convergence needs to be in order to have effective learning. It is perhaps too much for an exhaustive study of results with different number of time steps, but perhaps the authors could discuss where both classification and learning might break down due to insufficient time steps simulated and thus accumulated errors.

Lastly, there are a few related works that are worth citing, including Li & Pehlevan 2020 and Mancoo et al. 2020, both of which appeared in neurips last year. These papers describe spiking network models that do evolve to equilibrium states, defined by an implicit function (in those cases the implicit function happens to be a constrained optimization problem). They do not scale them up to deep learning problems, but they have already explored spiking networks evolving to equilibrium states.


**Time Spent Reviewing:**

6h (together with a postdoc)

---

> ### Author Response · Authors · 2021-08-10
> **Response to Reviewer W1bo**
>
> Thank you very much for appreciating our work and viewing our work novel and interesting. We respond to your valuable comments as follows.
>
> 1. About illustration of the model details. Thank you for pointing out this, our submission omitted it due to the space limit, and we will add an illustration figure in the revision.
>
> 2. About the network responding to input over time. Since our networks have thousands to hundreds of thousands of neurons, it is hard to demonstrate the responding of each neuron. We have already studied the overall network condition as shown in Figure 1 in Section 5.4 and the figures in Section E.2, i.e. we calculate the difference norm of the current (weighted) average firing rate on the equilibrium fixed-point equation at each time step. It demonstrates the convergent response of the network to the equilibrium state over time.
>
> 3. About the number of spikes. We calculate the total average firing rate and those of different layers for the model with AlexNet-F structure trained on CIFAR-10 with 30 time steps (both for IF and LIF model), and we present the average firing rate in the following (average over the whole dataset):
> | Layer | Average firing rate (IF model) | Average firing rate (LIF model) |
> | :----: | :----: | :----: |
> | layer1 | 0.0345 | 0.0166 |
> | layer2 | 0.0041 | 0.0039 |
> | layer3 | 0.0025 | 0.0024 |
> | layer4 | 0.0008 | 0.0008 |
> | layer5 | 0.0399 | 0.0177 |
> | total  | 0.0119 | 0.0066 |
>
>     The results demonstrate the firing sparsity of our trained model, as the average firing rate is only around or less than 1%. And it is interesting to show that the LIF model has a sparser response compared with the IF model. We note that TSSL-BP [44] also reported the statistics about firing rate, and according to their results, their trained model on CIFAR-10 has a roughly total 9.86% firing rate within 5 time steps. So it is interesting to find that our trained models have fewer spikes than theirs, even if we have more time steps than them (30 vs. 5), not to mention that our trained models have fewer neurons. It demonstrates the energy efficiency of our trained models and great potential for the application in neuromorphic hardware. We will add a discussion about this in the revision.
>
>     Also, we compare the average firing rate of the same model trained by our method and STBP [39, 40] (our implementation), and the results below show that our spikes are sparser as well. (CIFAR-10, AlexNet-F, LIF model, 30 time steps)
> | Layer | Average firing rate (STBP) | Average firing rate (IDE, ours) |
> | :----: | :----: | :----: |
> | layer1 | 0.0190 | 0.0166 |
> | layer2 | 0.0082 | 0.0039 |
> | layer3 | 0.0113 | 0.0024 |
> | layer4 | 0.0055 | 0.0008 |
> | layer5 | 0.0108 | 0.0177 |
> | total  | 0.0102 | 0.0066 |
>
> 4. About the stability of the equilibrium state during learning. It is always stable because, in practice, we restrict the spectral norm of weights to satisfy the assumption in the theorems and propositions by re-parameterizing and restricting the weight, as explained in Section D.1. And in the theorems and propositions, we have proven that if the spectral norm of weights satisfies the certain condition, the average firing rate will stably converge to only one equilibrium point.
>
> 5. About how good this convergence needs to be. The convergence condition w.r.t. time steps is illustrated in Sections 5.4 and E.2. And we further calculate the performance of a pretrained model under different time steps and present it here (AlexNet-F, CIFAR-10, LIF model, the original time step is 30):
> | Time steps | Accuracy | Total average firing rate |
> | :----: | :----: | :----: |
> | 5 | 10.67% | 0.0016 |
> | 10 | 74.39% | 0.0046 |
> | 15 | 87.07% | 0.0059 |
> | 20 | 90.16% | 0.0063 |
> | 25 | 91.33% | 0.0065 |
> | 30 | 91.82% | 0.0066 |
>
>     It shows that the accuracy will gradually decrease as the time step decreases, and when the time step is 5, the classification totally fails. We also briefly analyze the total average firing rate under these conditions as above, and it shows that the firing rate is very low when the time step is small. So the accuracy drop may also be partly due to the insufficient spikes. We will include this analysis in the revised version.
>
>     And for the case when learning might break, we tried to train and test the above model with only 5 time steps, and the result is:
> | Time steps | Accuracy |
> | :----: | :----: |
> | 30 | 91.74±0.09% |
> | 5 | 83.09% |
>
>     The training does not fail when the number of time steps is 5, but there would be a significant performance drop, and the accuracy would decrease and fluctuate in the latter part of training. It is probably because the gradient calculated by implicit differentiation could still be a descent direction though not exact, and in the latter part, it may not be a descent direction so the accuracy cannot be further improved.
>
> 6. About citation. Thank you very much for pointing out these papers. Li and Pehlevan (2020) model the excitation-inhibition balance for spiking networks as the minimax optimum of a constrained minimax optimization problem. They consider equilibrium from another perspective and do not guide the training of the network based on it. Mancoo et al. (2020) understand the computation of spiking networks as solving a constrained convex optimization problem and propose a learning method based on the neural hyperplanes, which is applicable to networks with a single hidden layer. Differently, our work considers equilibrium states with a fixed-point equation, based on which implicit differentiation can be easily applied to train networks for promising results, and is also scalable to multi-layer structure and deep learning problems. We will add this discussion in the related work section.
>
> Besides, there is no broader impact statement this year but limitation and social impacts, which we have included in the appendix Section F.

---

### Official Review · Reviewer_RLvt · 2021-07-18

**Rating:** 7
**Confidence:** 3

**Summary:**

This paper extends prior work [3,4] termed Implicit Differentiation on the Equilibrium state (IDE) to spiking neural networks. The authors derive a learning rule for the weights that is a product of two local equilibrium points in 2 phases. They perform experiments to compare their method with other SOTA SNNs, outperforming some.

**Limitations And Societal Impact:**

Yes

**Main Review:**

The authors work is both an extension of previous work [3,4] and a generalization of equilibrium propagation [34]. The key contribution is the extension to spiking neural networks and finding a seemingly local learning rule for the weights. Of course, key issues in biological plausibility remain, which the authors briefly allude to, namely requiring two phases to compute the equilibria $a^*$ and $\beta^*$, how the masking would take place, how $\frac{\partial \mathcal{L}}{\partial a^*}$ would be computed and fed in, etc. These are an intriguing possibilities that require further work. Their network + training outperforms SOTA SNNs in some benchmarks with less neurons / params and time steps. While this work is incremental, the paper could be an important step in improving and incorporating on-chip neuromorphic learning, given the emerging importance of energy-efficient neuromorphic computing and the possibilities this paper raises for online local learning. While the mathematical extensions to spiking neural networks are worked out, these latter aspects of local / bio-plausible / neuromorphic learning are not fleshed out, which greatly limits the applicability of the work. Overall, the work itself is clearly presented with some promising aspects, though raising further questions as above.

**Time Spent Reviewing:**

2.5

---

> ### Author Response · Authors · 2021-08-10
> **Response to Reviewer RLvt**
>
> Thank you very much for appreciating our work. Our work introduces equilibrium states for FSNNs and proposes a novel training method based on implicit differentiation, which avoids common training problems for SNNs and shares some interesting properties related to local learning rules. It will truly pave a solid way for improving on-chip neuromorphic learning and there could be many future works regarding different aspects. Still, we respond to your valuable comments as follows.
>
> 1. First, we would like to emphasize the contribution of our work. The training for SNNs is important for both computer science and neuroscience communities, and while the biological plausibility requires further work, the contribution to computer science is not simply an incremental extension. Our novel training method tackles the common difficulties for SNN training including non-differentiability and large memory costs, and models can be trained on GPUs with high precision and low latency. Though our work is inspired by deep equilibrium models [3,4], it is not simply incremental due to two important differences between our work and theirs.
>     - The first is that our work is a novel framework for SNN training rather than constructing implicit models defined by an equation as them, who replace both forward and backward computation with implicit root-finding methods. It broadens the application of implicit differentiation for equilibrium states which are solved by explicit SNN computation instead of high-precision root-finding methods.
>     - The second is that deriving equilibrium states for FSNNs is not a trivial extension of implicit models. Implicit models start from the explicit ANN computation as $x^{t+1} = f_{\theta}(x^t)$, so it is easy to consider the equilibrium with the equation $x=f_{\theta}(x)$. However, for SNNs there are no trivial convergent explicitly iterative equations, and we make efforts to consider properties of SNNs to introduce equilibriums states w.r.t (weighted) average firing rates for (leaky) integrate and fire models with convergence guarantee.
>
>     As for the equilibrium propagation [34], our work has a very different derivation and method compared with them, while happens to have a similar two-stage property and connection to the Hebbian rule. It is not incremental generalization as well, and there could also be future work to study the connections.
>
> 2. For the issues about biological plausibility and potential further works, it is true that our work only briefly discusses some interesting properties, and many further efforts could be spent to flesh out more details. Our work introduces the new idea of the equilibrium states w.r.t. (weighted) average firing rates and the novel training method based on implicit differentiation for SNNs, which we hope would build a basic foundation for future works regarding both bio-plausible learning and local neuromorphic learning.
>
> 3. As for the application of the work, while we admit that the current work cannot perform local neuromorphic learning to train SNNs, we provide an alternative approach that models can be trained on GPUs and then applied on neuromorphic hardware for energy-efficient inference applications. As shown in the experiments, our model can achieve better performance with fewer neurons and parameters within a small number of time steps, and meanwhile, the spikes are sparse as well (we provide some statistics of the firing rate as below. CIFAR-10 dataset, AlexNet-F structure, 30 time steps).
> | Layer | Average firing rate (IF model) | Average firing rate (LIF model) |
> | :----: | :----: | :----: |
> | layer1 | 0.0345 | 0.0166 |
> | layer2 | 0.0041 | 0.0039 |
> | layer3 | 0.0025 | 0.0024 |
> | layer4 | 0.0008 | 0.0008 |
> | layer5 | 0.0399 | 0.0177 |
> | total  | 0.0119 | 0.0066 |
>
>     It shows that the firing of neurons is extremely sparse, and our trained models have fewer spikes compared with previous work such as TSSL-BP [44] whose firing rate is roughly 9.86%. We will add a discussion of this in the revision.
>     So our models trained on GPUs can be directly applied on neuromorphic hardware with high energy efficiency and high performance. This application is still promising, and the future work for local neuromorphic learning would be further promising.

---

### Decision · Program_Chairs · 2021-09-27

**Decision:**

Accept (Spotlight)

**Comment:**

The paper proposes a novel method to train spiking neural networks (SNNs) by considering the equilibrium state of their firing rates. They exploit results on implicit differentiation to compute gradients in the network. They show that the method achieves state-of-the-art performance on several data sets, including CIFAR-10 and CIFAR-100.
The method is based on rigorous proofs about the equilibrium states of an SNN.

All reviewers agree that the paper presents interesting results. The manuscript is well-written and the experimental results are convincing.